# Delineating the early dissemination mechanisms of acral melanoma by integrating single-cell and spatial transcriptomic analyses

Chuanyuan Wei [1,2,8], Wei Sun[3,4,8], Kangjie Shen[1,2,8], Jingqin Zhong[3,4], Wanlin Liu[3,4], Zixu Gao[1,2], Yu Xu[3,4], Lu Wang[1,2], Tu Hu[3,4], Ming Ren[1,2], Yinlam Li[1,2], Yu Zhu[1,2], Shaoluan Zheng[5], Ming Zhu[1,2], Rongkui Luo [6], Yanwen Yang[1,2], Yingyong Hou [6], Fazhi Qi[1,2], Yuhong Zhou[7] ✉, Yong Chen [3,4] ✉ & Jianying Gu [1,2,5] ✉

Acral melanoma (AM) is a rare subtype of melanoma characterized by a high incidence of lymph node (LN) metastasis, a critical factor in tumor dissemination and therapeutic decision-making. Here, we employ single-cell and spatial transcriptomic analyses to investigate the dynamic evolution of early AM dissemination. Our findings reveal substantial inter- and intra-tumor heterogeneity in AM, alongside a highly immunosuppressive tumor microenvironment and complex intercellular communication networks, particularly in patients with LN metastasis. Notably, we identify a strong association between *MYC*+ Melanoma (*MYC*+MEL) and *FGFBP2*+NKT cells with LN metastasis. Furthermore, we demonstrate that LN metastasis requires a metabolic shift towards fatty acid oxidation (FAO) induced by MITF in *MYC*+MEL cells. Etomoxir, a clinically approved FAO inhibitor, can effectively suppress MITF-mediated LN metastasis. This comprehensive dataset enhances our understanding of LN metastasis in AM, and provides insights into the potential therapeutic targeting for the management of early AM dissemination.

Acral melanoma (AM) is a rare subtype of melanoma that occurs mainly in the sun-shielded skin of the palms, soles, and nail beds[1]. Although only accounting for 2–3% of all melanoma cases, half of Asian, African, and Hispanic patients have AM[2]. Unlike the extensive somatic mutations associated with ultraviolet signatures in cutaneous melanoma (CM), AM displays a lower incidence of activated mutations, such as those in *BRAF* and *NRAS*, while possesses a higher frequency of mutations in *KIT*, *NF1*, and *PTEN*[3]. AM is also characterized by high copy number variation (CNV), including *CDK4* and *CCND1* amplification, *TP53* inactivation, and *TERT* alteration[4,5]. A recent study suggests[6] that late-arising focal amplifications in cytoband 22q11.21, especially in *LZTR1*, are associated with lymph node

[1]Department of Plastic and Reconstructive Surgery, Zhongshan Hospital, Fudan University, Shanghai 200032, P. R. China. [2]Cancer center, Zhongshan Hospital, Fudan University, Shanghai 200032, P. R. China. [3]Department of Musculoskeletal Oncology, Fudan University Shanghai Cancer Center, Shanghai 200032, P. R. China. [4]Department of Oncology, Shanghai Medical College, Fudan University, Shanghai 200032, P. R. China. [5]Department of Plastic and Reconstructive Surgery, Xiamen Branch of Zhongshan Hospital, Fudan University, Xiamen 361015, P. R. China. [6]Department of Pathology, Zhongshan Hospital, Fudan University, Shanghai 200032, P. R. China. [7]Department of Medical Oncology, Zhongshan Hospital, Fudan University, Shanghai 200032, P. R. China. [8]These authors contributed equally: Chuanyuan Wei, Wei Sun, Kangjie Shen. ✉e-mail: zhou.yuhong@zs-hospital.sh.cn; chenyong@fudan.edu.cn; prof_jianyinggu@163.com

(LN) involvement and distant metastasis of AM. Compared with CM, AM is characterized by a severe immunosuppressive state, fewer effector/cytotoxic CD8⁺T cells, natural killer (NK) cells and a near-complete absence of γδ T cells, while is enriched with regulatory T cells (Tregs), and exhausted CD8⁺T cells[7,8]. From a clinical perspective, AM often presents at a more advanced stage and has a worse prognosis[9]. However, AM is treated in a manner similar to CM currently. For targeted therapy, *BRAF*-mutant AM shows a similar response rate to BRAF inhibitors as *BRAF*-mutant CM. Unfortunately, only 19% of AM patients harbor *BRAF* mutation[3]. For immune checkpoint blockade (ICB) therapy targeting anti-PD-1 and/or anti-CTLA4, the response rates for AM, ranging from 15-20%, are lower than those for CM[10]. Therefore, our understanding of AM remains limited, underscoring the pressing demand for the formulation of therapeutic regimens rooted in its distinctive biological characteristics.

LN metastasis is a foothold for further tumor dissemination and predicts cancer recurrence and poor prognosis[11]. LN metastases resist T cell-induced cytotoxicity, induce the activation of antigen-specific Tregs, and develop tumor-specific immune tolerance, subsequently facilitating distant tumor colonization[12]. LN status is closely related to the clinical stage of melanoma; patients without LN metastasis are diagnosed with stage I/II disease, but once LN metastasis occurs, they are diagnosed with stage III[13]. Accordingly, the five-year survival rate for localized melanomas can reach 99%, whereas regional and metastatic melanomas have poorer prognoses, with five-year survival rates of 68% and 30%, respectively[14]. In addition, intraoperative completion lymph node dissection and postoperative adjuvant treatment are recommended for LN metastatic melanomas, especially AM, because of the deeper Breslow thicknesses and higher positive rates of sentinel LN (ranging from 28 to 30%)[15]. In summary, LN metastasis predicts tumor progression and often guides therapeutic schedules for melanoma.

To date, numerous studies have been devoted to investigate the molecular mechanisms underlying LN metastasis in melanoma. A comparative analysis of paired primary and LN-metastatic tumors showed that LN metastasis requires tumor cells to undergo a metabolic shift toward fatty acid oxidation (FAO)[16]. The yes-associated protein pathway is selectively activated in LN metastatic tumors, leading to the upregulation of genes in the FAO pathway. Macrophages located in the sub-capsular sinus produce pro-tumoral IL-1α after recognition of tumoral antigens and promote melanoma metastasis to the sentinel LN via the IL-1α/STAT3 axis[17]. A markedly immunotolerant tumor microenvironment (TME) has been observed in melanoma-bearing LN, with reduced and impaired NK cells and increased *CD57⁺PD-1⁺CD8⁺*T cells, leading to compromised anti-melanoma immunity and a high relapse rate[18]. However, these results are mainly obtained by comparing primary and LN metastatic lesions in the CM. The distinction between primary melanoma lesions in LN metastatic patients and non-metastatic patients is still unclear, especially in AM patients.

In this study, we perform single-cell RNA sequencing (scRNA-seq) and spatial transcriptome sequencing (ST-seq) to systematically investigate the heterogeneity and ecosystem of primary tumor tissues in AM patients with LN metastasis and without metastasis. The aim is to identify the potential functions of various cellular components during early tumor dissemination in AM. Our findings reveal substantial inter- and intra-tumor heterogeneity in AM, alongside a highly immunosuppressive TME and complex intercellular communication networks, particularly in patients with LN metastasis. The *MYC*⁺ Melanoma (*MYC*⁺MEL) and *FGFBP2*⁺NKT subclusters are tightly correlated with LN metastasis and poor prognosis. LN metastasis requires that melanoma cells to undergo a metabolic shift towards FAO induced by MITF, a key transcription factor in *MYC*⁺MEL cells. Local administration of Etomoxir, a clinically approved FAO inhibitor, suppresses MITF-mediated LN metastasis. We also test these findings through in vivo experiments,

and multiplex immunohistochemistry (IHC) assays, and verify these results using internal and external data from a large number of clinical samples.

## Results

### Single-cell transcriptome atlas of AM

To comprehensively characterize the ecosystem of AM, we collected the primary tumor tissues from 12 patients for scRNA-seq, including six had LN metastasis (LN⁺AM) and six did not (LN⁻AM) (Fig. 1a). The clinicopathological characteristics of these patients are presented in Supplementary Table 1, and representative tumor H&E images are shown in Supplementary Fig. 1.

An aggregate gene expression matrix with 52,382 cells and 30,097 genes was generated. 38 distinctive cell clusters were obtained with a resolution of 2.0, and visualized by the Uniform Manifold Approximation and Projection (UMAP) plot (Fig. 1b and Supplementary Fig. 2a). To identify the main cell types, each cluster was annotated using the cluster-specific marker genes[7,8,19] (Supplementary Fig. 2b, c). Eleven clusters were annotated as tumor cells because they expressed high levels of melanoma-associated marker genes (*MLANA*, *MITF*, *PRAME* and *SOX10*). The remaining clusters were divided into 12 non-tumor cell types. Supporting the supervised cell type-specific marker analysis, an unsupervised global clustering similarity matrix was used to classify these 38 clusters into their corresponding meta-clusters (Fig. 1c). We confirmed that each cell type expressed its well-known marker genes with high specificity (Fig. 1d, e). Meanwhile, the highest level of CNV, which plays an important role in the pathogenesis and poor prognosis of tumor patients[20], was observed in melanoma cells (Supplementary Figs. 2d and 3a–c).

These 13 cell types were detected in almost every patient; however, their proportions varied greatly (Fig. 1f and Supplementary Fig. 4a–e). Melanoma was the most abundant cell type with the highest transcript and CNV levels, which was consistent with its highly malignant characteristics (Fig. 1f). T cells were the most abundant type of immune cells, and this is consistent with previous research results[19,21]. Compared to that in LN⁻AM, plasmacytoid dendritic cells (pDC) and NK cells were slightly decreased in LN⁺AM, whereas no difference was observed in T cells between the two groups (Fig. 1f). The stromal compartment, which encompasses endothelial cells, fibroblasts, and epidermal cells, exhibited reduced prominence within the TME of LN⁺AM, potentially at the expense of tumor cells (Fig. 1f). Collectively, AM exhibits high heterogeneity, and the assessment of immune-cold characteristics cannot be simplified by looking solely at the proportion of total immune cells.

### Spatial transcriptome suggests a "cold" TME

Spatial information is critical for understanding tumor biology; unfortunately, it is missing from scRNA-seq data[22]. Here, we performed ST-seq to acquire in situ gene expression profiles of three patients with LN⁺AM and two with LN⁻AM, and the results of quality control are presented in Supplementary Fig. 5a–c. The spots were divided into tumor and non-tumor regions using the conditional autoregression-based deconvolution (CARD) algorithm[23] (Fig. 2a). In the tumor regions, we obtained six clusters with significant differences in CNV levels, and these clusters constituted spots for different patients. For instance, patient 6 was associated with cluster 2 and 4, whereas patient 11 was associated with cluster 1 with the highest CNV levels (Fig. 2b, c, and Supplementary Fig. 6a, b). In the non-tumor regions, five clusters were obtained, and cluster 4 was shared by patients 6, 11, 14 and 15 (Fig. 2d, e, and Supplementary Fig. 6c, d). These results indicate high inter-tumor heterogeneity in tumor regions.

We further performed a deconvolution analysis to determine cell proportions using the CARD[23] and robust cell-type decomposition (RCTD)[24,25] methods. We presented the matched scRNA-seq and ST-seq data of patient 6 and 11 (Supplementary Fig. 7a, b), and showed that the

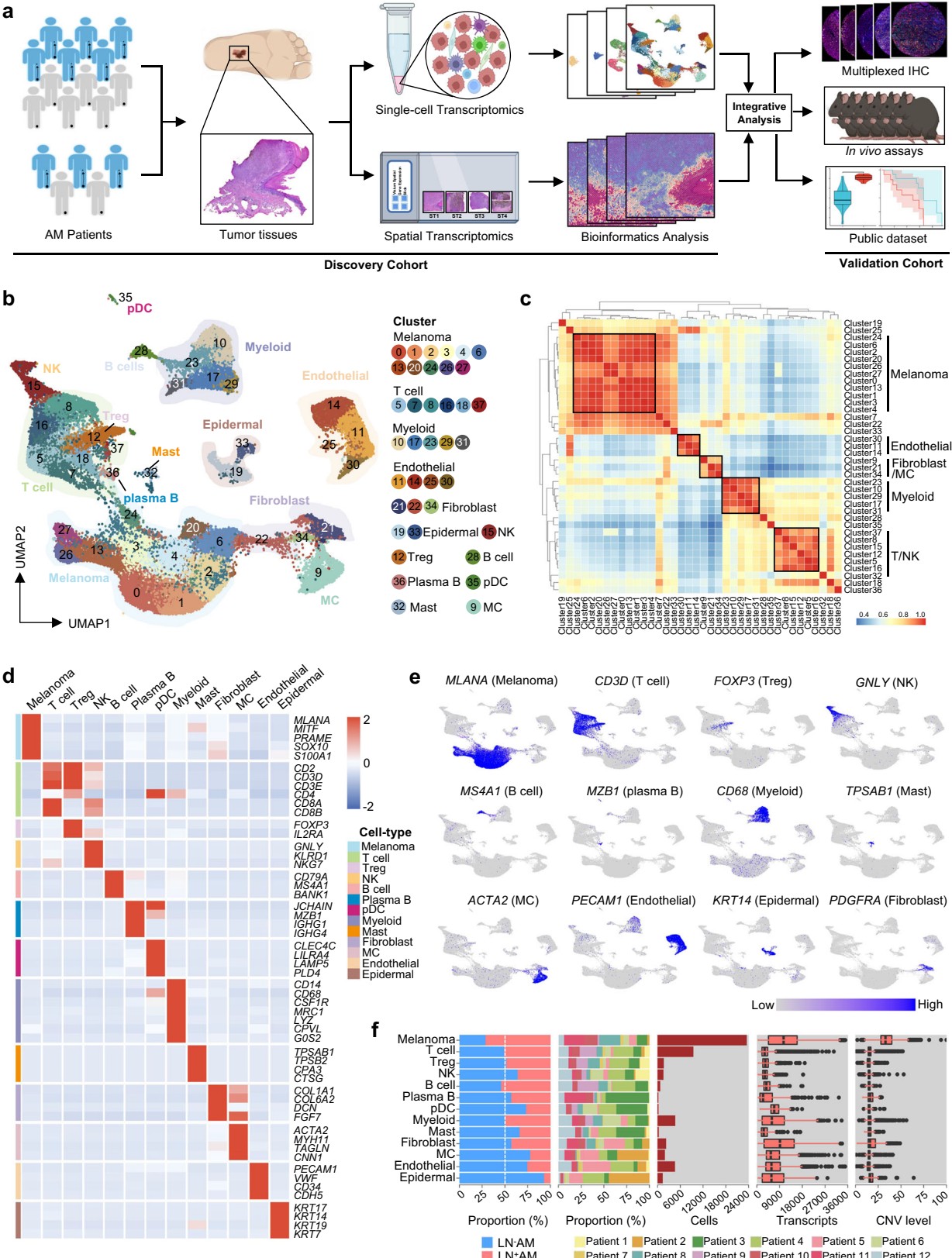

percentages of most cell types in ST-data were consistent with that in scRNA-seq. However, a minor difference might be attributed to the different regions obtained for sequencing. Immune cells mainly gathered in non-tumor areas but were obviously decreased in tumor areas (Fig. 2f, and Supplementary Fig. 8a). Notably, fewer immune cells were infiltrated in the tumor ecosystem of LN⁺AM than that of LN⁻AM

(Fig. 2g, and Supplementary Fig. 8b). Hence, AM presents a "cold" TME, especially in patients with LN metastases.

**High heterogeneity of melanoma cells at a single-cell level**
To decipher the landscape of tumor cells, 23,501 melanoma cells were regrouped into five subclusters. These subclusters had unique gene

**Fig. 1 | Single-cell characterization of the AM ecosystem. a** Schematic representation of the study design. AM patients with and without LN metastasis were enrolled, and their primary tumor lesions were collected for scRNA-seq and ST-seq analysis. In vivo experiments, multiplex IHC images, and external datasets were utilized for results validation. A part of the image has been adapted from Biorender.com. **b** UMAP plot demonstrating the cell distribution from 12 primary AM tissues, color-coded by the annotated cell types. **c** Heatmap depicting pairwise correlations among 38 clusters derived from 12 tumor tissues. Clustering identified five coherent expression programs across tumors. **d** Heatmap identifying the expression of selected marker genes in the annotated cell types. Source data are provided in the Source Data file. **e** Feature plots presenting classical marker genes for the annotated cell types. **f** Bar plot indicating the fraction of annotated cell types originating from patients with LN⁺AM and LN⁻AM, as well as from each individual patient; histogram displaying the cell count of the annotated cell types; box plots indicating the median (middle line), 25th and 75th percentiles (box), and outliers (individual points) of transcripts and CNV levels in the annotated cell types. Source data are provided in the Source Data file.

expression patterns and biological functions, and their proportions varied greatly among patients, indicating high intra- and inter-tumor heterogeneity (Fig. 3a, b, and Supplementary Fig. 9a, b). *MYC*⁺MEL cells were increased in LN⁺AM and expressed high levels of *MYC*, *MITF*, *SNAI2*, and *KIT*, all of which play crucial roles in tumor progression. For example, MYC regulates downstream target genes primarily involved in proliferation, differentiation, metabolism, and angiogenesis and is considered an attractive therapeutic target[26]. Accordingly, *MYC*⁺MEL cells were functionally enriched in mesenchyme and stem cell development (Fig. 3b). The *CXCL10*⁺MEL subcluster, with high levels of *CXCL10*, *CXCL8* and *IRF1*, was functionally enriched in regulation of immune response, immune effector process, and leukocyte chemotaxis. The *TMSB4X*⁺MEL subcluster with high levels of *TMSB4X*, *IGKC* and *IGLC2*, was functionally enriched in regulation of i-κB kinase/NF-κB signaling, cytokine secretion, and cell chemotaxis (Fig. 3b). These cancer immunity-related clusters were downregulated in LN⁺AM, suggesting that tumor cells could promote LN dissemination by shaping a "cold" TME (Fig. 3a). The *CENPF*⁺MEL subcluster, with high levels of *CENPF*, *CCNB1*, *PCNA*, and *MKI67*, was functionally enriched in DNA replication and cell cycle, whereas the *NEAT1*⁺MEL subcluster with high expression of *NEAT1*, *KLF6*, *JUN* and *FOS*, was functionally enriched in response to reactive oxygen species and oxidative stress (Fig. 3b).

We then delineated the characteristics of the five melanoma subclusters. The CNV levels of total melanoma cells, as well as each subcluster, were significantly higher in LN⁺AM than that in LN⁻AM (Fig. 3c and Supplementary Fig. 9c, d). We then applied SCENIC analysis[27] to these subclusters (Fig. 3d). Transcription factors (TFs) associated with tumor metastasis and progression, such as *MITF*, *MYC*, and *CTNNB1*, showed high transcription activity in the *MYC*⁺MEL subcluster. Similarly, immune/inflammation-related TFs, such as *NFKB2*, *STAT1*, and *RELA*, and *FOXO3*, showed high activity in *CXCL10*⁺MEL and *TMSB4X*⁺MEL subclusters, which underlies the immunomodulatory phenotype of tumor cells. For example, a transcription rheostat orchestrated by RELA confers T cells with the innate ability to produce IFN-I/III[28]. *E2F1*, *POLE4* and *TFDP1* are known TFs associated with cell proliferation, and they were mainly enriched in *CENPF*⁺MEL cells. *FOSB*, *JUN*B, and *JUN* were highly activated in the *NEAT1*⁺MEL subcluster, and contained activator protein-1 (AP-1), which is widely involved in various tumor events including differentiation, proliferation, and apoptosis[29].

We then depicted the pseudo-time trajectory of these melanoma subclusters (Fig. 3e, f, and Supplementary Fig. 10a–c). The phase 0 was predominated by the *CENPF*⁺MEL subcluster (85.1%) and expressed high levels of proliferation-associated genes, such as *CCNB1* and *CENPF*. Accordingly, RNA velocity confirmed that the melanoma cells were originated from *CENPF*⁺MEL cells (Supplementary Fig. 10d). Phase 1 primarily composed *CXCL10*⁺MEL and *TMSB4X*⁺MEL subclusters (65.3%), and expressed high levels of immune-related molecules, such as *CCL2*, *C1R* and *STAT3* (Fig. 3g). Phase 2 primarily composed *MYC*⁺MEL and *NEAT1*⁺MEL subclusters (70.5%), and expressed high levels of metastasis-related molecules, such as *MITF*, *KIT*, and *VIM* (Fig. 3g). These results highlight the multifaceted roles played by tumor cells within the AM ecosystem.

## Activated fatty acid metabolic pathway in *MYC*⁺MEL cells

Compared to LN⁻AM, the *MYC*⁺MEL cluster was significantly increased in LN⁺AM, suggesting a crucial role in LN metastasis (Fig. 4a). We then mapped the gene signatures of *MYC*⁺MEL subcluster using a public dataset containing 26 primary AM samples[6]. The *MYC*⁺MEL subcluster was significantly increased in LN⁺AM, and high *MYC*⁺MEL score was associated with poorer prognoses than that of low score (Supplementary Fig. 11a, b). Notably, we found that multiple FAO-related genes, including *EPHX1*, *GAPDHS*, and *HSP90AA1*, were elevated in the *MYC*⁺MEL subcluster (Fig. 4b). Accordingly, gene sets associated with fatty acid metabolism were enriched in the *MYC*⁺MEL subcluster (Fig. 4c). Considering that LN metastasis necessitates a metabolic shift towards FAO[16], we speculated that the hyperactivation of the FAO pathway observed in the *MYC*⁺MEL cells could potentially contribute to LN metastasis.

## MITF promotes LN metastasis via the FAO pathway

To identify the key factors promoting FAO in the *MYC*⁺MEL subcluster, we performed correlation analyses using the scRNA-seq data (Supplementary Fig. 11c, d). Surprisingly, we found a positive correlation between MITF and its regulon activity and the FAO score. Using the ST-seq data, we observed a good spatial consistency between *MITF* expression and the FAO score, and higher FAO scores were presented in MITF^High spots (Fig. 4d, e). These results indicate that MITF may promote FAO activation in AM.

To explore the potential role of the MITF-mediated FAO activation in LN metastasis, we conducted a mouse footpad model (Fig. 4f, and Supplementary Fig. 11e). *Mitf*-overexpressing B16F0 (B16F0-*Mitf*) and B16F0-Vector cells were subcutaneously implanted into the footpad regions of C57BL/6 mice. We extended the timeline to achieve a higher LN metastasis rate, especially considering the lower LN metastasis rate after drug treatment. At the endpoint, we observed that the size and metastatic area of the popliteal LN were significantly larger in the B16F0-*Mitf* group than in the B16F0-Vector group (Fig. 4g, h, and Supplementary Fig. 11f, g). We further applied Etomoxir, a clinically approved FAO inhibitor, topically on the anterolateral side of the legs of mice[30]. Although Etomoxir treatment did not influence the size of the primary tumor or the weight of the tumor-bearing mice, it markedly inhibited LN metastasis (Fig. 4g, h, and Supplementary Fig. 11g). All of these findings collectively support that MITF contributes to increased FAO activity, thereby promoting LN metastasis in AM.

## Functional impairment of the antitumor immunity in LN⁺AM

A total of 13,521 immune cells were extracted and regrouped into 18 subclusters (Fig. 5a–c). The proportions of these subclusters varied greatly among patients and between the LN⁺AM and LN⁻AM groups (Supplementary Fig. 12a–c). CD8⁺T cells were designated as *GZMK*⁺ (effector), *ANXA1*⁺ (memory), *IFNG*⁺ (cytotoxic), and *CXCL13*⁺ (exhausted) cells according to cluster-specific marker genes. Compared to LN⁻AM, *CXCL13*⁺CD8⁺T cells were increased, whereas *IFNG*⁺CD8⁺T cells were slightly decreased in LN⁺AM, indicating a compromised antitumor TME (Fig. 5a). For CD4⁺T cells, *CXCL13*⁺ (exhausted) and *FTH1*⁺ (naive) cells were slightly increased in LN⁺AM, whereas no differences

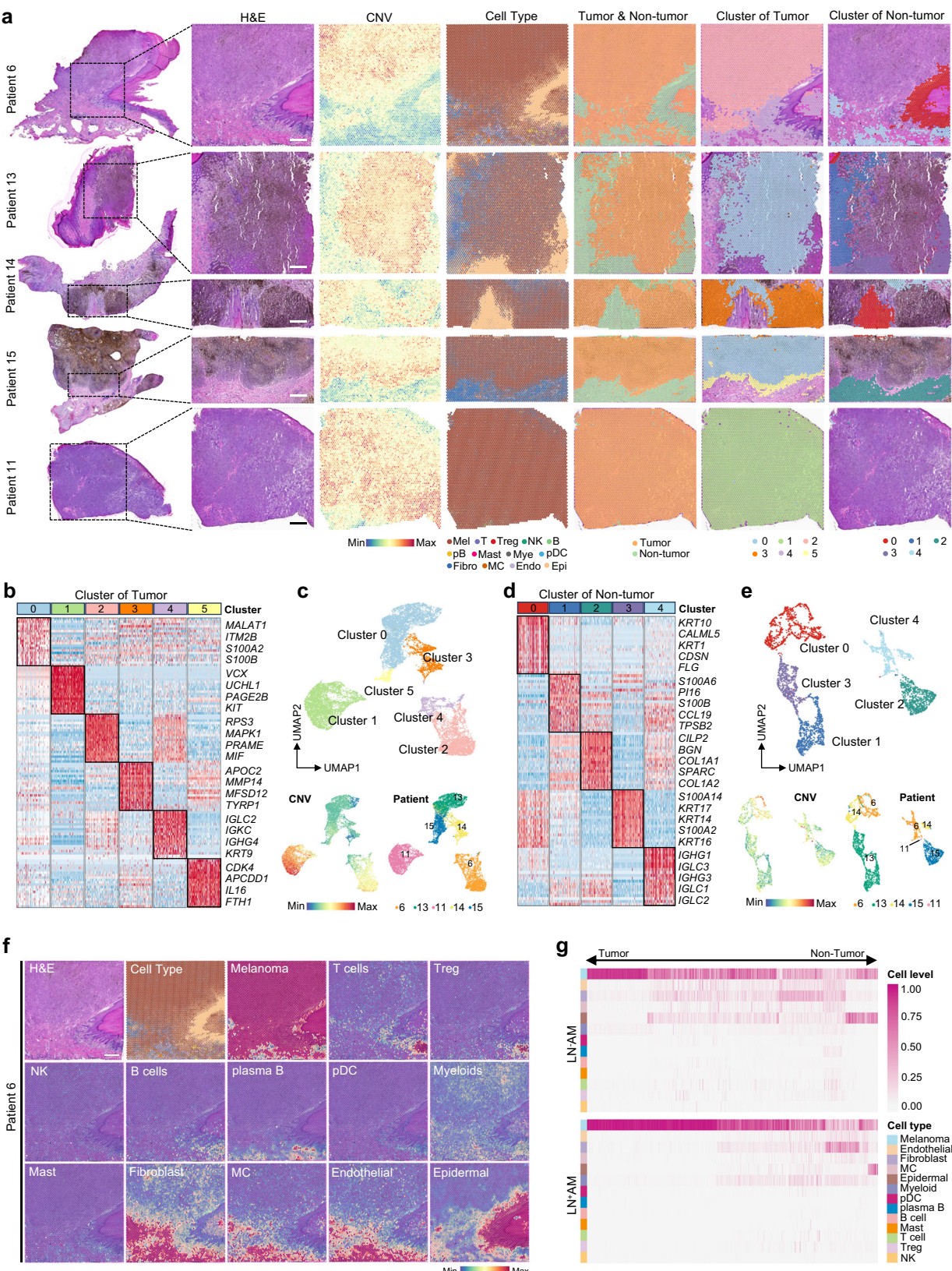

were observed in the proportions of *KLRB1*+ (effector) and *FOS*+ (memory) cells between LN+AM and LN-AM (Fig. 5a). NK/NKT cells exhibit potent anti-tumor responses[31]. Here, *XCL2*+NK and *FGFBP2*+NKT cells were decreased in LN+AM than in LN-AM (Fig. 5a). These results suggest that the antitumor immunity of LN+AM patients is obviously weakened.

To decipher these subclusters, we performed the GSVA analysis and observed that multiple immunoregulatory pathways, including interferon-α/γ response and IL2/STAT5 signaling, were enriched in CD8+T subclusters (Fig. 5d). Interestingly, almost all the selected pathways were activated in *XCL2*+NK and *FGFBP2*+NKT cells, indicating that they had high proliferative and antitumor activities (Fig. 5d). Next,

**Fig. 2 | Spatial transcriptome suggests a "cold" TME in AM. a** The first column displays H&E-stained images showing the tumor tissues from five AM patients, with the dotted boxes indicating the selected regions for ST-seq. In the second column, each spot contains an average of 1–10 cells, colored according to CNV levels using Seurat. In the third to sixth columns, spots are colored by the annotated cell types in scRNA-seq data, tumor or non-tumor regions, defined six clusters in tumor regions, and defined five clusters in non-tumor regions using the CARD method. Scale bar, 1000 μm. **b** Heatmap depicting cluster-specific genes in tumor regions. Source data are provided in the Source Data file. **c** UMAP plot showing spots in tumor regions, colored based on annotated clusters, CNV levels, and samples. **d** Heatmap illustrating cluster-specific genes in non-tumor regions. Source data are provided in the Source Data file. **e** UMAP plot showing spots in non-tumor regions, colored based on annotated clusters, CNV levels, and samples. **f** Spatial feature plots of annotated cell types from scRNA-seq data in five AM tissue sections with similar results, and representative images were presented. Scale bar, 1000 μm. **g** Heatmap showing the infiltration levels of annotated cell types in LN⁻AM (upper panel) and LN⁺AM (lower panel) patients from tumor to non-tumor regions. Source data are provided in the Source Data file.

we presented the selected genes to describe their biological functions (Fig. 5e). For example, *CXCL13*⁺CD8⁺T and *CXCL13*⁺CD4⁺T cells, designated as exhausted T cells, expressed high levels of immune checkpoint molecules such as *TIGIT*, *CTLA4*, *LAG3*, and *PDCD1*. Importantly, *FGFBP2*⁺NKT cells expressed the highest levels of antitumor cytokines, indicating their potent antitumor effects. We also used SCENIC analysis and identified a set of TFs implicated in the biology of different immune cell types (Supplementary Fig. 12d). In *FGFBP2*⁺NKT cells, multiple TFs were highly activated, especially *CEBPB*, which is associated with an active TME and favorable prognosis of metastatic melanomas[32]. Additionally, we further performed RNA velocity analysis to investigate the ongoing processes of these subclusters (Fig. 5f). T cells mainly originated from naive cells, differentiated towards memory, effector, and cytotoxic cells, and ended with exhausted cells. Exhausted CD8⁺T cells were primarily derived from effector CD8⁺T cells, whereas cytotoxic CD8⁺T cells were derived from both effector and memory CD8⁺T cells. Importantly, *FGFBP2*⁺NKT cells mainly originated from effector CD8⁺T cells, rather than *XCL2*⁺NK and CD4⁺T cells.

**Decreased *FGFBP2*⁺NKT cells accelerate LN metastasis**
Given their powerful tumor-killing abilities, CD8⁺T and NK cells were extracted for further analysis. Using published signatures of cytotoxicity and exhaustion[33,34], we observed that *CXCL13*⁺CD8⁺T and *FGFBP2*⁺NKT cells had the highest exhaustion and cytotoxicity scores, respectively (Fig. 5g). Accordingly, *FGFBP2*⁺NKT cells expressed the highest levels of cytotoxic genes, such as *GZMB*, *GZMH*, *PRF1*, and *GNLY*, indicating that *FGFBP2*⁺NKT cells had the strongest tumor-killing effects (Fig. 5h). Additionally, we tested several immune checkpoint molecules and found that *PDCD1* and *CTLA4* were expressed at low levels, while *LAG3* and *TIGIT* were highly expressed in CD8⁺T and NK cells (Supplementary Fig. 12e). This suggests that ICB therapy targeting LAG3 and TIGIT may potentially yield more potent antitumor effects in AM.

Owing to the potent anti-tumor ability, we further investigated the relationship between *FGFBP2*⁺NKT cells and LN metastasis in AM. *FGFBP2*⁺NKT cells were significantly decreased in LN⁺AM compared with those in LN⁻AM (Fig. 5i). This result was validated by a public dataset[6], namely, the infiltrated score of *FGFBP2*⁺NKT cells was significantly lower in LN⁺AM than that of LN⁻AM, and patients with low *FGFBP2*⁺NKT scores had worse prognoses (Supplementary Fig. 12f, g). These results indicated that decreased *FGFBP2*⁺NKT cells may participate in LN metastasis. Using multiplex IHC assays, we showed that *FGFBP2*⁺NKT cells (positive for CD8, NCAM1, and FGFBP2, but not for CD4) were mainly present in the para-tumor regions and were sparsely distributed in the tumor regions (Fig. 5j). Statistical analysis revealed that *FGFBP2*⁺NKT cells were significantly decreased in the TME of LN⁺AM compared to that of LN⁻AM, and patients with decreased *FGFBP2*⁺NKT cells tended to have worse prognoses (Fig. 5k). These results indicate that decreased *FGFBP2*⁺NKT cells are closely correlated with LN metastasis in AM.

**Macrophages exert anti-inflammatory effects in AM**
For myeloid cells, a total of 5212 cells were collected and reallocated into 13 subclusters, including macrophages, monocytes, neutrophils,

DC, and pDC (Fig. 6a–c). For example, *SPP1*⁺ Macrophages (*SPP1*⁺Mac) expressed high levels of *SPP1*, and *CCL2* and *MMP9* levels were elevated in this subcluster (Fig. 6c). Compared with LN⁻AM, more macrophages, especially *CXCL10*⁺Mac cells (26.62% versus 19.35%), and fewer *GZMB*⁺pDC cells (3.78% versus 8.65%), were present in LN⁺AM (Fig. 6d and Supplementary Fig. 13a–c). These tumor-associated macrophages varied greatly in their signaling pathways and metabolic characteristics (Fig. 6e). *SPP1*⁺Mac and *APOE*⁺Mac cells showed increased levels of reactive oxygen species (ROS), fatty acid metabolism, glycolysis, and oxidative phosphorylation, whereas *CXCL10*⁺Mac and *F13A1*⁺Mac cells were enriched in the complement and interferon-α response. Metabolism associated pathways, such as bile acid metabolism and heme metabolism, were activated in almost all macrophages. For monocytes and neutrophils, multiple immune/inflammatory pathways were enriched, such as TNF-α signaling via NFKB, IL6/JAK/STAT3 signaling, and INF-γ response, suggesting key roles in immune/inflammatory regulation. Using SCENIC analysis, we identified distinct transcriptional activities in these subclusters. For instance, *SOX4* and *IRF7* were highly activated in *GZMB*⁺pDC (Supplementary Fig. 13d).

We performed RNA velocity analysis to investigate this process in myeloid cells (Fig. 6f). Macrophages were mainly derived from *APOE*⁺Mac and then differentiated towards two directions: *SPP1*⁺Mac, and *CXCL10*⁺Mac and *F13A1*⁺Mac. Using Monocle2, we confirmed that the pseudo-time trajectory was initiated with *APOE*⁺Mac, followed by differentiation into *SPP1*⁺Mac and *CXCL10*⁺Mac, and ending with *F13A1*⁺Mac (Fig. 6g). Accordingly, *APOE*⁺Mac showed low expression of inflammatory/immune-related genes, and *SPP1*⁺Mac, *CXCL10*⁺Mac and *F13A1*⁺Mac expressed high levels of inflammatory/immune-related genes. However, the expression profiles were quite different, suggesting that different biological functions were performed (Fig. 6h). We also observed that these subclusters expressed high levels of immune checkpoint molecules; for example, *SPP1*⁺Mac expressed high levels of *CD274* and *HAVCR2*, *CXCL10*⁺Mac expressed high levels of *CD274*, *PDCD1*, and *TIGIT*, and *F13A1*⁺Mac expressed high levels of most immune checkpoint molecules, which was consistent with the terminal differentiation status (Fig. 6h). By calculating the M1/M2 polarization scores[35], we found that most macrophages were skewed towards the M2 phenotype, except for *APOE*⁺Mac cells, which presented low M1/M2 scores (Fig. 6i, j), indicating that most macrophages exerted anti-inflammatory effects in the AM ecosystem.

**Cancer-associated fibroblasts promote angiogenesis**
A total of 10,148 stromal cells were redistributed into 13 subclusters designated as cancer-associated fibroblasts (CAFs), muscle cells (MCs), epidermal cells, and endothelial cells (Fig. 7a, b). CAFs expressed high levels of *MMP2*, *CFD*, and *FN1* and were divided into two different types. *MMP2*⁺CAFs, *CFD*⁺CAFs and *SOD2*⁺CAFs subclusters expressed high levels of cytokines and chemokines, similar to inflammatory CAFs (iCAFs, PDGFRA⁺)[36]. *THY1*⁺CAFs, which expressed high levels of *THY1*, *ACTA2*, and *TAGLN*, similar to MCs, was designated as myoCAFs. We found that these iCAFs subclusters were upregulated, and myoCAFs and MCs subclusters were decreased in LN⁺AM (Fig. 7c, d and Supplementary Fig. 14a–d). Based on the expression of *RGCC*, *ACKR1*, *SEMA3G*, and *PROX1*, the endothelial cells were divided into capillary, venous, arterial, and lymphatic endothelial cells (Fig. 7a, b). Using

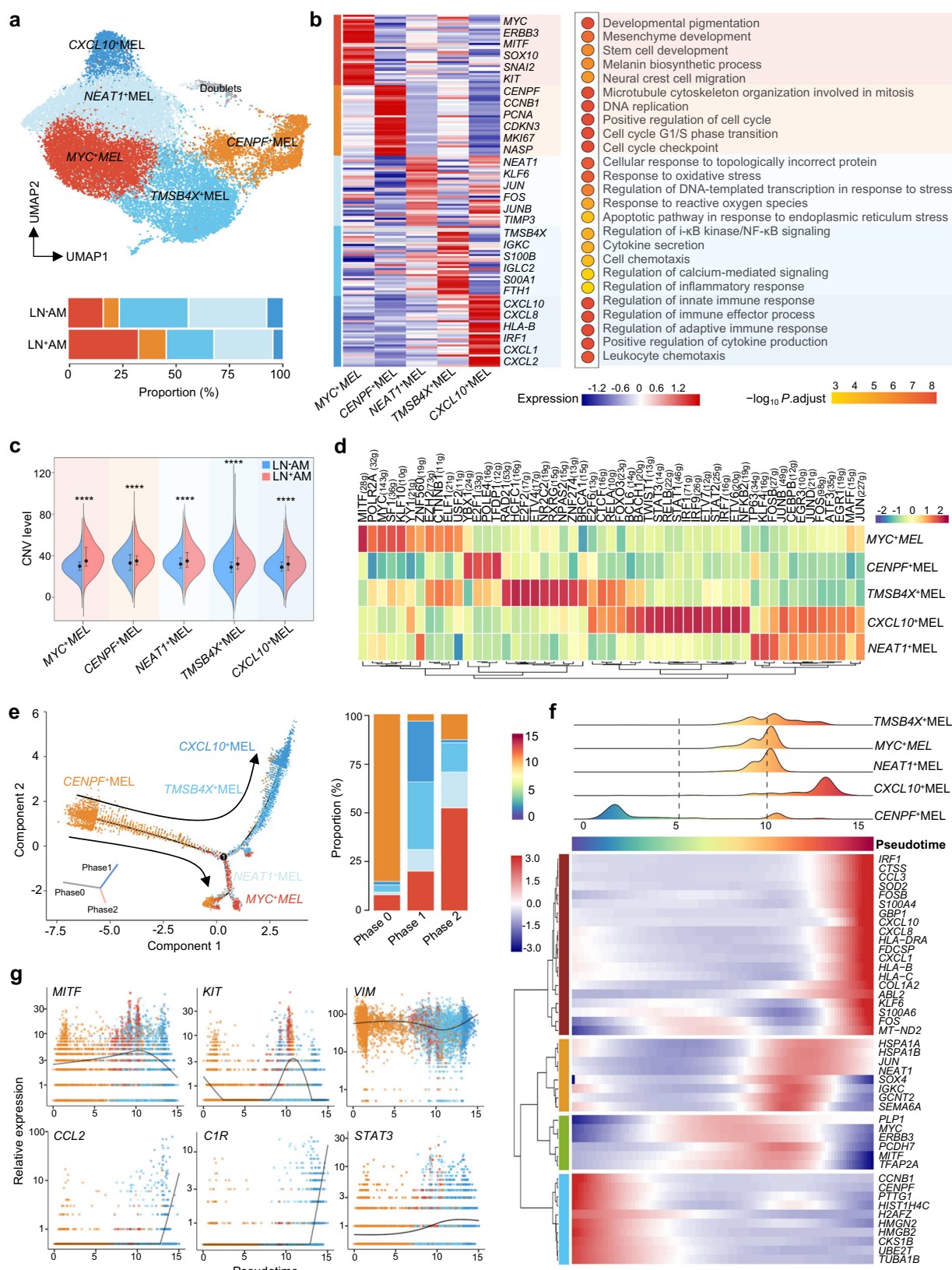

SCIENIC analysis, we identified crucial TFs that may mediate the biological functions of these subclusters, for example, IRF8 was highly activated in *CCL21*⁺ endothelial (*CCL21*⁺Endo) cells (Supplementary Fig. 14e).

Functional enrichment analysis revealed that different pathways were activated in stromal cells (Fig. 7e). Multiple pathways, such as

angiogenesis, epithelial-mesenchymal transition, and myogenesis, were enriched in CAFs and MCs. These subclusters also have distinct biological functions. For example, immune/inflammatory pathways, such as cell chemotaxis to FGF and G-CSF production, were enriched in iCAFs. Negative regulation of myoblast proliferation, and regulation of cardiac muscle tissue development were enriched in MCs, while

**Fig. 3 | Functions and evolutionary trajectories of distinct AM subclusters.**
**a** UMAP plot of all melanoma cells color-coded by annotated subclusters, and a bar plot showing the proportion of these annotated subclusters originating from LN$^+$AM and LN$^-$AM patients. Source data are provided in the Source Data file.
**b** Heatmap (left) displaying cluster-specific genes, and dot plot showing (right) the corresponding enriched biological functions of these annotated melanoma subclusters. Significance was determined using a two-sided Wilcoxon rank-sum test. Source data are provided in the Source Data file. **c** Split violin plot illustrating the CNV levels of annotated subclusters colored by LN$^-$AM and LN$^+$AM. ****$P$ < 0.0001; two-sided Mann–Whitney $U$-test; $MYC^+$MEL, LN$^-$AM $n$ = 1268, LN$^+$AM $n$ = 6635, $P$ = 2.9e-128; $CENFP^+$MEL, LN$^-$AM $n$ = 501, LN$^+$AM $n$ = 2,211, $P$ = 2.9e-9; $NEAT1^+$MEL, LN$^-$AM $n$ = 1429, LN$^+$AM $n$ = 2580, $P$ = 9.7e-20; $TMSB4X^+$MEL, LN$^-$AM $n$ = 2004, LN$^+$AM $n$ = 2553, $P$ = 2.7e-41; $CXCL10^+$MEL, LN$^-$AM $n$ = 1432, LN$^+$AM $n$ = 2549, $P$ = 1.3e-

35. The back circle indicates the median, and the bottom and top lines indicate the first and third quartiles, respectively. Source data are provided in the Source Data file. **d** Heatmap showing the expression of key transcription factors (estimated using SCENIC) among the annotated melanoma subclusters. **e** Pseudotime-ordered analysis (left) of these AM cells identified two distinct cell fates, colored by subclusters. The arrow indicates the potential evolutionary direction in the trajectory. Bar plot (right) showing the proportions of each subcluster in the different branches of the trajectory. Source data are provided in the Source Data file. **f** Heatmap displaying the dynamic changes in gene expression along the pseudotime (lower panel). Subclusters are labeled by colors (upper panel). **g** Two-dimensional plots showing the dynamic expression of selected genes along the pseudo-time, colored by subclusters.

myoCAFs exerted partial roles of iCAFs and MCs (Supplementary Fig. 14f). Compared with myoCAFs and MCs, iCAFs expressed higher levels of *MMP2*, *MMP14*, *FAP* and *PDGFRA* (Supplementary Fig. 14g). RNA velocity analysis revealed that these CAFs subclusters were mainly derived from *MMP2*$^+$CAFs and then differentiated towards myoCAFs and MCs (Fig. 7f).

Notably, the angiogenesis pathway was enriched in all CAFs subclusters (Fig. 7e). Using ST-seq data, we found that endothelial cells and fibroblasts were significantly colocalized, presenting good spatial consistency (Fig. 7g). Pearson's correlation analysis showed a positive correlation between the scores of endothelial cells and CAFs, and importantly, a higher correlation coefficient was observed in the LN$^+$AM group (Fig. 7h). We further speculated that CAFs promoted angiogenesis and formed fibrovascular niches, which play a positive role in tumor progression[37]. To confirm this, we performed multiplex IHC assays and found that CAFs often presented together with endothelial cells and wrapped around them to form fibrovascular niches (Fig. 7i). Compared with LN$^-$AM, more fibrovascular niches were present in LN$^+$AM. These results show that CAFs promote angiogenesis and form fibrovascular niches to promote the early dissemination of AM.

## Complex intercellular communication in AM

The intercellular communication is extensive and complex in AM ecosystems detected by the CellChat analysis (Fig. 8a and Supplementary Fig. 15a, b). Melanoma and stromal cells mainly served as senders, as they had high outgoing interaction strength, whereas immune cells mainly served as receivers, as they had high incoming interaction strength (Fig. 8b). Plasma B cells showed less communication as they had low incoming and outgoing interaction strength. Notably, we found that the interaction number and strength were slightly higher in the ecosystem of LN$^+$AM than in that of LN$^-$AM, however, no statistical difference was noted (Supplementary Fig. 15c, d).

The overall incoming and outgoing signaling patterns are presented (Supplementary Fig. 16a, b). We showed that the CD99 and MIF signals were activated in almost all cell types, indicating that they played extensive roles. The CCL and CXCL signals were mainly sent by stromal cells and received by diverse cell types. The CCL signaling was received by endothelial cells, indicating a role in angiogenesis, while the CXCL signaling was also received by immune cells and played a role in remodeling the TME. The CLEC signaling was sent by multiple cell types, and mainly received by NK cells, indicating that the activity and cytotoxicity of NK cells was regulated via the CLEC pathway. Compared to LN$^-$AM, the immune/inflammatory pathways, such as the CCL, CXCL, and CLEC signals, were decreased in LN$^+$AM, which may be related to a weaker TME. The ANGPTL, APP, COLLAGEN, LAMININ, and MK signals were mainly sent by stromal cells, and received by multiple cell types, suggesting that stromal cells, especially CAFs, played extensive roles in the ecosystem.

The MIF pathway was activated in both the LN$^+$AM and LN$^-$AM groups (Supplementary Fig. 16c). MIF was highly expressed in almost all cell types and participated in both the innate and adaptive immune responses (Supplementary Fig. 16d). Mechanistically, MIF independently interacts with CD74 in a hetero-complex with CD44, CXCR2, CXCR4, and ACKR3 to initiate downstream MAPK and PI3K pathways, all of which influence tumor initiation, growth, and metastatic dissemination[38]. In the MK pathway, midkine (MDK) encodes a small family of secreted growth factors that promote cell growth, migration, and angiogenesis during tumorigenesis[39]. Notably, compared with LN$^-$AM, the MK pathway was activated in LN$^+$AM. MDK was primarily secreted by melanoma cells and interacted with multiple cell types in LN$^+$AM, whereas it was secreted by fibroblasts and endothelial cells in LN$^-$AM (Supplementary Fig. 16e). Among these receptors, NCL and ITGB1 were highly expressed in most cell types (Supplementary Fig. 16f). Using the stlearn method, we confirmed the spatial interaction of ligand-receptor (L-R) pairs in the MK pathway (Supplementary Fig. 16g).

The CLEC network primarily comprised CLEC2C/CLEC2B/ CLEC2D-KLRB1 pairs and was specifically expressed in non-tumor cells (Supplementary Fig. 17a). Compared to LN$^-$AM, activation of the CLEC pathway was significantly decreased in LN$^+$AM (Fig. 8c). The genes of the C-type lectin superfamily are highly expressed in NK cells and may be involved in the regulation of the activity of NK cells[40]. The decreased activity of the CLEC pathway in LN$^+$AM might result in a decrease in NK and NKT cells. We also presented the GALECTIN signaling pathway network, which was significantly activated in LN$^+$AM (Fig. 8d). Galectins play a central role in tumorigenic processes by delivering regulatory signals that contribute to a variety of cellular events leading to tumor cell proliferation, metastasis, angiogenesis and immune escape[41,42]. LGALS9, the main ligand, mainly expressed in myeloid cells in LN$^-$AM, whereas it also expressed in endothelial cells in LN$^+$AM. CD44, CD45, and HAVCR2 acted as receptors, indicating an important role of endothelial cell-derived LAGALS9 in the LN metastasis of tumor cells (Fig. 8e, and Supplementary Fig. 17b, c). Importantly, we confirmed the crosstalk between the L-R pairs of the GALECTIN pathway in our AM cohort (Fig. 8f). In summary, our results reveal that specific intercellular communication has the potential to shape the unique TME of AM.

## Discussion

Compared with CM, AM typically has a higher rate of LN metastasis, which is a foothold for further tumor dissemination and recurrence[43]. Here, we performed scRNA-seq and ST-seq analyses to detect the tumor/immune transcriptome landscape of primary AM tissues with and without LN metastasis at the spatial and temporal levels. Our study identified that AM exhibited strong inter- and intra-tumor heterogeneity, a highly inhibitory TME, and complex cell-cell communications, especially in LN$^+$AM. $MYC^+$MEL and $FGFBP2^+$NKT subclusters were closely associated with LN metastasis and

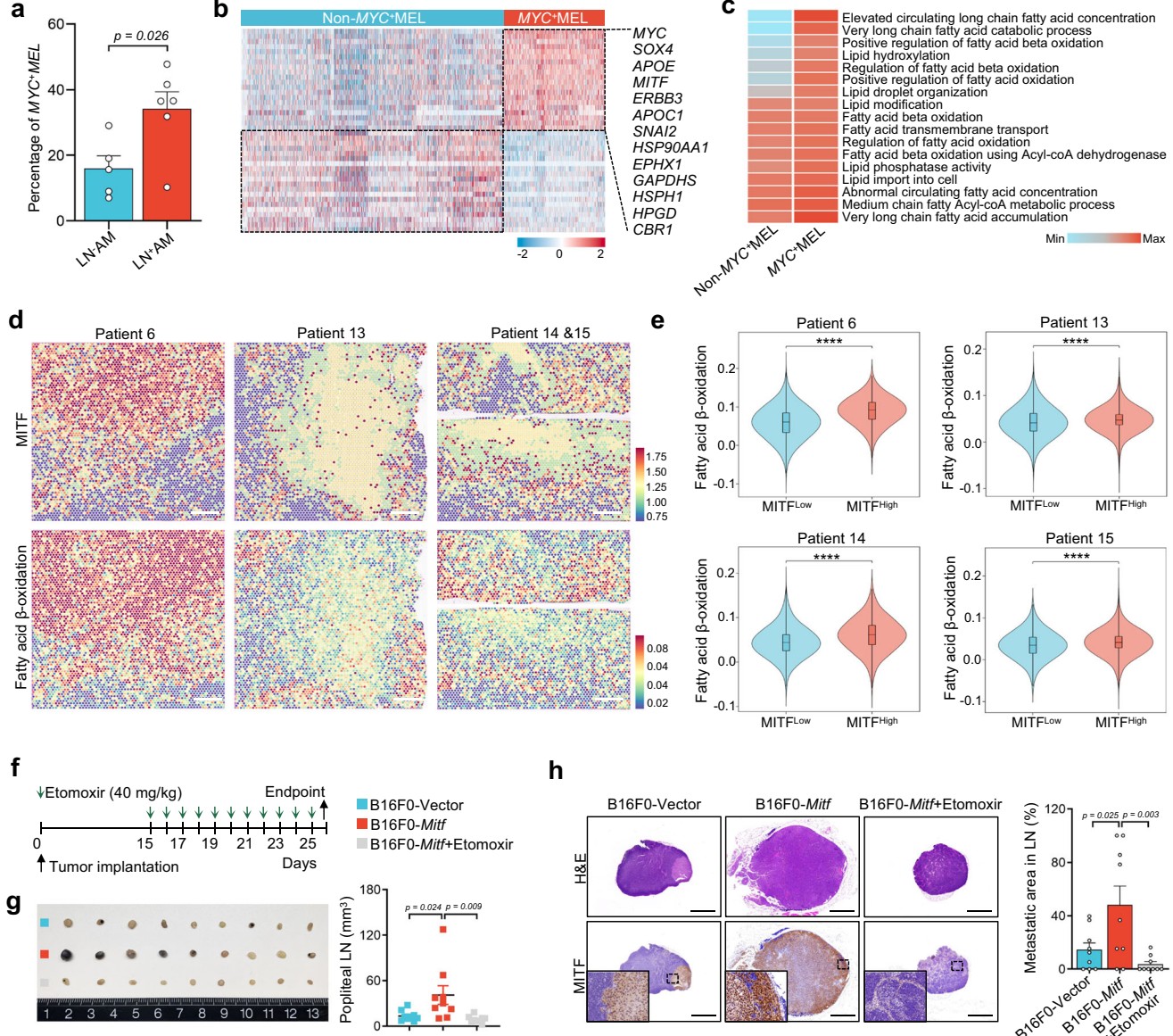

**Fig. 4 | MITF-mediated FAO activation drives tumor LN metastasis. a** Histogram illustrating the proportion of *MYC*+MEL cells in LN−AM (*n* = 5) and LN+AM (*n* = 6) groups. Significance was determined using a two-sided unpaired student's *t*-test. Error bars represent the mean ± SD. Source data are provided in the Source Data file. **b** Heatmap displaying expression levels of selected DEGs in *MYC*+MEL versus non-*MYC*+MEL groups using scRNA-seq data. **c** Heatmap showing fatty acid metabolism-related pathways in *MYC*+MEL and non-*MYC*+MEL groups. Source data are provided in the Source Data file. **d** Spatial feature plots showing *MITF* expression and FAO score in selected tissue sections. Scale bar, 1000 μm. **e** Violin plots displaying the FAO score between MITF^High and MITF^Low spots in each tissue section. ****P*-value < 0.0001; two-sided Mann–Whitney *U*-test; patient 6 ($n^{Low}$ = 2495, $n^{High}$ = 2495): *P* = 7.2e-175; patient 13 ($n^{Low}$ = 2374, $n^{High}$ = 2373): *P* = 7.2e-15; patient 14 ($n^{Low}$ = 989, $n^{High}$ = 989): *P* = 1.0e-28; patient 15 ($n^{Low}$ = 1444, $n^{High}$ = 1444): *P* = 7.9e-11.

Box center lines, bounds of the box, and whiskers indicate medians, first and third quartiles, and minimum and maximum values within 1.5× IQR of the box limits, respectively. Source data are provided in the Source Data file. **f** Treatment schedule for Etomoxir in the tumor footpad implantation model of C57BL/6 mice. **g** Image of the gross appearances of popliteal LNs among the indicated groups, and a scatter plot showing the sizes of popliteal LNs in B16F0-Vector, B16F0-*Mitf*, and B16F0-*Mitf*+Etomoxir groups (*n* = 9). Significance was determined using a one-way ANOVA test. Error bars represent the means ± SD. Source data are provided in the Source Data file. **h** H&E and anti-MITF stained images of popliteal LNs in the indicated groups (*n* = 9). The metastatic area in the total LN was calculated using the histogram. Significance was determined using a one-way ANOVA test. Error bars represent the mean ± SD. Scale bar, 1000 μm. The experiment was repeated once with similar results. Source data are provided in the Source Data file.

poor prognosis in AM patients. Mechanistically, LN metastasis required tumor cells to undergo a metabolic shift towards fatty acid metabolism induced by the key transcription factor MITF in *MYC*+MEL cells. Etomoxir could suppress LN metastasis by targeting the FAO pathway (Fig. 9).

We have expanded our discussion to acknowledge the high inter- and intra-tumor heterogeneity observed in AM, as previously demonstrated in various studies[8]. These studies classified melanoma cells into distinct subgroups based on different criteria. For instance,

in the TCGA-SKCM cohort[44], melanoma cells were categorized into immune, keratin, or MITF-low subgroups. Another study classified melanoma cells as immune, normal-like, pigmented, or proliferative subtypes[45]. Belote et al. highlighted the loss of melanocyte differentiation markers during melanoma progression, and the proportion of melanocytes that had readopted a neonatal-like signature was associated with worse prognosis[46]. Recently, melanoma cells were divided into five orthogonal functional cell clusters that were involved in TGF-β signaling, Type I interferon, WNT

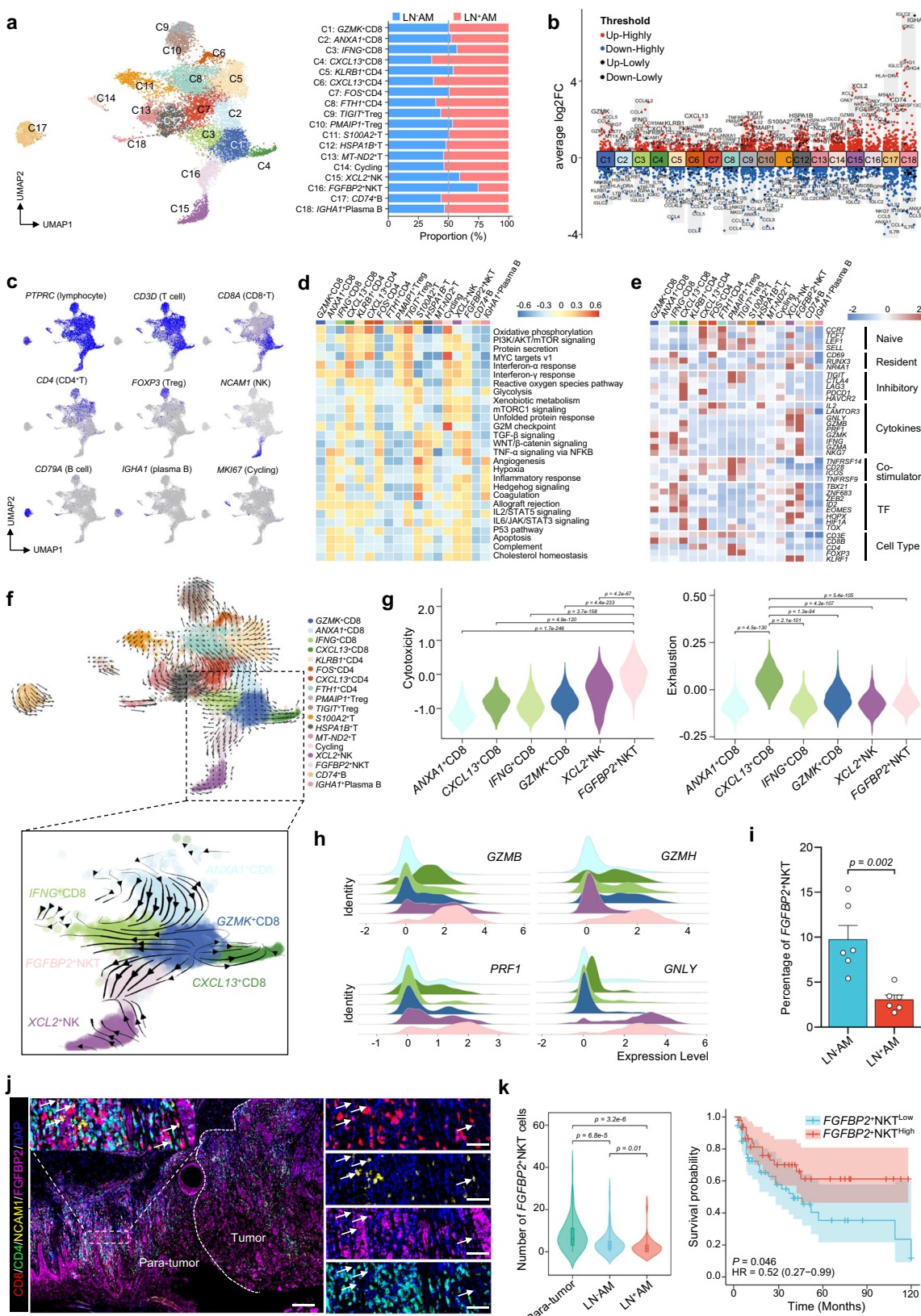

signaling, cell cycle, and cholesterol efflux signaling[8]. Here, our study mainly focused on AM and investigated the differences in tumor cells based on their LN metastatic status. AM cells were regrouped into five subclusters with distinct biological functions. They were initialized with the proliferative subcluster (*CENPF*+MEL) and then differentiated towards two directions: tumor metastasis and stemness, and immune/

inflammatory regulation. Among them, the *MYC*+MEL subcluster exhibited functional enrichment in mesenchymal-like malignant clusters in CM, which are closely associated with tumor invasion and metastasis. These results indicate that tumor cells can benefit from early dissemination by altering their metastatic and immunoregulatory abilities.

**Fig. 5 | Decreased *FGFBP2*⁺NKT cells in LN⁺AM. a** UMAP plot (left) illustrating immune cells, color-coded by associated subclusters, and a bar plot (right) displaying the proportion of each subcluster in the LN⁺AM and LN⁻AM groups. Source data are provided in the Source Data file. **b** Volcano plot indicating DEGs in the annotated subclusters of immune cells. Source data are provided in the Source Data file. **c** Feature plots of classical marker genes used for subcluster annotation. **d** GSVA analysis of selected hallmark pathways in these subclusters. **e** Heatmap indicating the expression of selected gene sets (including naive, resident, inhibitory, cytokines, co-stimulatory, transcription factors (TF), and cell type) in these subclusters. **f** RNA velocity analysis demonstrating the evolutionary trajectory of these subclusters. The CD8⁺T, NK and NKT subclusters are highlighted and enlarged in the lower panel. **g** Violin plots displaying the cytotoxicity and exhaustion scores in the selected subclusters. *****P*-value < 0.0001; Kruskal–Wallis rank-sum test was used. Source data are provided in the Source Data file. **h** Ridge plots showing the expression levels of *GZMB*, *GZMH*, *PRF1*, and *GNLY* in these subclusters. **i** Histogram illustrating the proportion of *FGFBP2*⁺NKT cells in LN⁺AM (*n* = 6) and

LN⁻AM (*n* = 6) groups using our scRNA-seq data. Significance was determined using an unpaired two-sided student's *t*-test, *P*-value = 0.002. Error bars represent means ± SD. Source data are provided in the Source Data file. **j** Representative image of *FGFBP2*⁺NKT cells (positive for CD8, NCAM1, and FGFBP2) in AM cohort (*n* = 101) using the multiplex IHC assay. The white arrow indicates representative *FGFBP2*⁺NKT cells. The scale bars on the left and right are 200 μm and 50 μm, respectively. The experiment was repeated once with similar results. **k** Violin plot (left) showing the number of *FGFBP2*⁺NKT cells in Para-tumor (*n* = 72), LN⁻AM (*n* = 73), and LN⁺AM (*n* = 28) groups, with significance determined using a Kruskal–Wallis rank-sum test. Box center lines, bounds of the box, and whiskers indicate medians, first and third quartiles, and minimum and maximum values within 1.5x IQR of the box limits, respectively. KM analysis (right) showing the overall survival rate of 101 AM patients with high and low levels of *FGFBP2*⁺NKT cells using the two-sided log-rank test (*P* = 0.046, hazard ratio (HR) = 0.52). Source data are provided in the Source Data file.

*MITF* is a "lineage-specific survival" oncogene that is essential for melanoma initiation, progression, and relapse[47]. *MITF* amplification was found to be more prevalent in metastatic disease and correlated with decreased overall survival rates in melanoma patients[47]. Low levels of *MITF* generate G1-arrested, invasive, and senescent cells, whereas cells expressing *MITF* either proliferate or differentiate[48]. MITF promotes the transformation of saturated fatty acids into monounsaturated fatty acids by regulating SCD1 expression, thereby promoting tumor cell proliferation[49]. Recently, our group identified compound TT-012, which dynamically binds to MITF and destroys the dimer formation and DNA-binding ability of MITF, inhibits the growth of melanoma cells with high MITF expression along with tumor growth and metastasis, indicating a crucial role of MITF in melanoma progression[50]. Together, we posit that the involvement of MITF in melanoma is both multifaceted and intricate, potentially assuming distinct functions contingent on the temporal and spatial context within the tumor environment. Here, we aimed to identify key subgroups that mediate early tumor metastasis, especially primary tumor cells that are about to metastasize but have not yet metastasized. In this intermediate state between primary and metastatic lesions, we demonstrated that MITF is involved in regulating the energy metabolism, and the overactivated FAO pathway promotes the metastatic potential of AM cells. Previous study has reported a substantial upregulation of the FAO pathway in LN metastasis lesions[16]. We further showed that the activation of the FAO pathway is not confined solely to LN metastasis tissues, but is already activated in primary tumor tissues with LN metastasis. Together, we propose the crucial role of the enhanced MITF-FAO axis in promoting LN metastasis, and blocking this axis may be a potential target for inhibiting tumor progression in AM.

Using scRNA-seq, Li and colleagues performed an in-depth analysis of the lymphocyte compartment of matched primary and metastatic melanoma samples and found that LN metastasis was associated with remarkably fewer CD8⁺T cells, NK cells, monocytes, and macrophages and showed increased infiltration of two subsets of CD4⁺T cells and B cells[7]. A markedly immunotolerant environment in melanoma-bearing sentinel LN was observed, as indicated by reduced and impaired NK cells and increased levels of *CD57*⁺*PD-1*⁺CD8⁺T cells, which are known to exhibit low tumor-killing capabilities[18]. Other changes observed in melanoma-bearing sentinel LN include (i) reduced *CD69*⁺CD8⁺T cell/Treg cell ratio, (ii) high PD-1 expression on CD4⁺T and CD8⁺T cells, and (iii) high CTLA-4 expression on γδ T cells. Here, no difference was observed in the proportion of total T lymphocytes between the LN⁺AM and LN⁻AM groups. While in detail, exhausted CD8⁺T and CD4⁺T cells were increased, and *XCL2*⁺NK and *FGFBP2*⁺NKT cells were decreased in LN⁺AM compared to those in LN⁻AM, indicating a compromised anti-tumor TME. In Zhang et al.'s study[8], NK cells (expressing high levels of *FGFBP2* and *KLRD1*) are

divided into six clusters in the TME of melanoma. In contrast to Zhang et al.'s study[8], which primarily focused on T cells, our research delved into NK cells and NKT cells. We showed that there were a group of cells located between NK and T cells in the UMAP plot, expressed markers of both NK and T cells, and had termed them *FGFBP2*⁺NKT cells due to their high FGFBP2 expression. Notably, our in-depth analysis revealed that these *FGFBP2*⁺NKT cells possessed the strongest tumor-killing ability within the AM ecosystem. Crucially, we had observed a significant decrease in the population of *FGFBP2*⁺NKT cells in tumors with LN metastasis, underscoring their potential relevance to the metastatic process. All of these suggest that the primary tumors mainly present a "cold" TME during LN metastasis.

The intercellular communication is complex in primary AM tumors. Overall, stromal and tumor cells mainly served as ligands that send signals, whereas immune cells served as receptors that receive signals. This phenomenon was more prominent in LN⁺AM, suggesting that the complex cell-cell communication was participated in the early dissemination of tumors. Chemerin, TGF-β, and IL-1 regulated the interaction of *FAP*⁺CAFs and *SPP1*⁺Mac, resulting in the formation of immune-excluded desmoplastic structure and limiting the T cell infiltration[51]. *CD36*⁺CAF-derived MIF potentiated the capacity of MDSCs to promote immunosuppressive TME and tumor stemness via IL-6/STAT3 activation[52]. Here, we showed that all CAFs clusters promoted angiogenesis in the TME and were always present as fibrovascular niches. More fibrovascular niches were observed in the LN⁺AM group, indicating that this structure played a positive role in early tumor dissemination.

This study has several limitations. First, the high heterogeneity among patients and limited sample size poses challenges in identifying common characteristics. However, to overcome the problems of heterogeneity and the small sample size, we conducted a dual verification of the results obtained using internal and external data. Second, in vitro validation of the crucial role of the MITF-FAO pathway in LN metastasis is lacking, owing to the lack of immortalized AM cell lines. To overcome this problem, we verified the results via in vivo animal experiments. Meanwhile, we are actively constructing AM cell lines, which will be tested in subsequent experiments. Third, our study lacks three-dimensional (3D) spatial analysis. The tumor grew in a 3D environment, and we presented only two-dimensional (2D) spatial information on the interaction between tumors and their TME. Currently, methods such as PASTE can be used to expand downstream analyses and cellular interaction networks defined by 3D spatial structures[53,54]. Last, our study did not include analysis of LN and other distant metastases, and we lacked in-depth analysis of MITF function in these metastatic lesions. In future work, we plan to comprehensively analyze the dynamic process of tumor dissemination from the primary lesion to lymph nodes and other organs. Nonetheless, our study has provided valuable insights into the intermediate

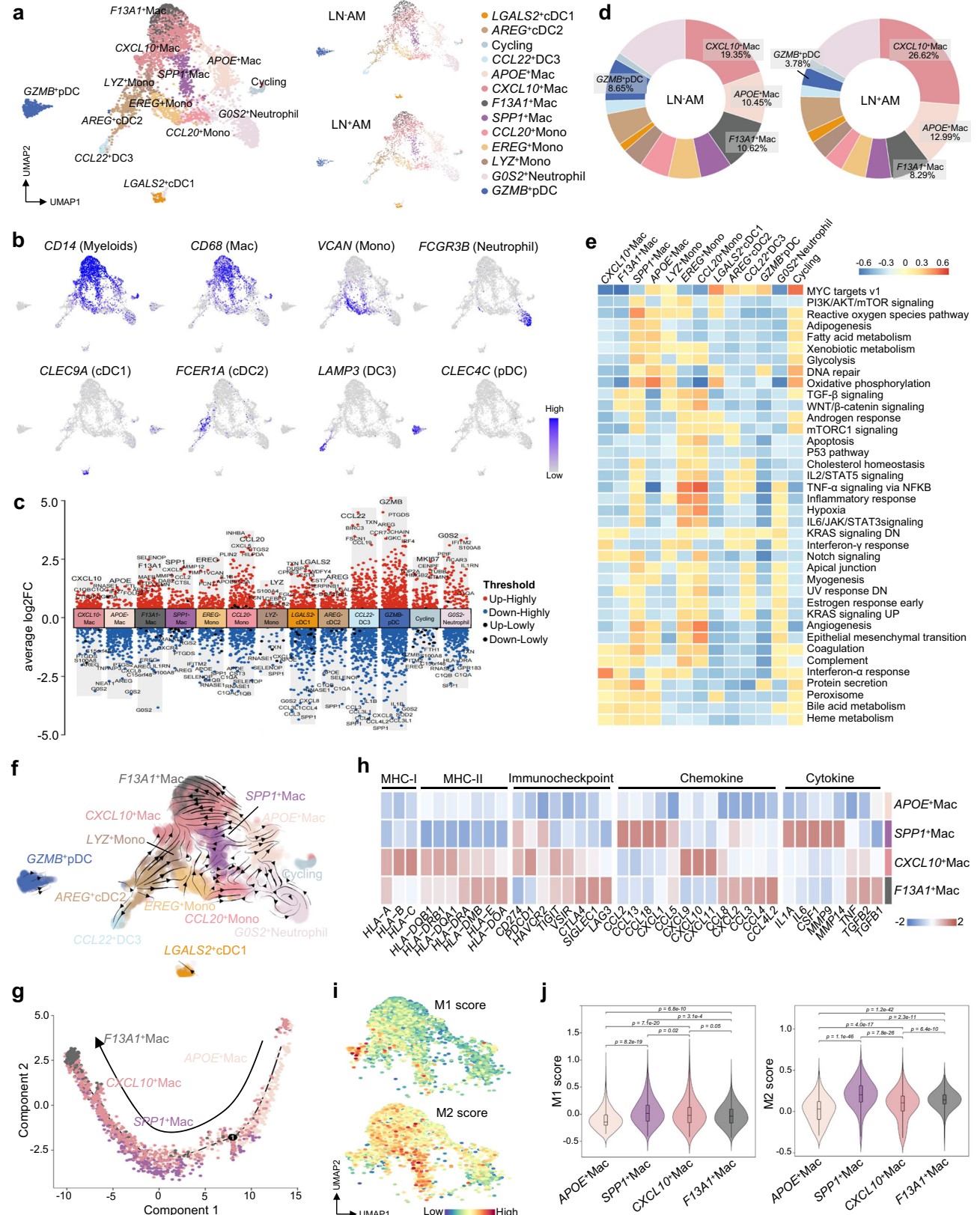

state of MITF function between primary and metastatic lesions, specifically how MITF can enhance metastatic potential by regulating energy metabolism.

In conclusion, this study provides a better understanding of the heterogeneity of tumor ecosystems between patients with LN+AM and LN−AM in terms of immune and tumor phenotypes. Our results can be a valuable resource, facilitating a deeper understanding of the mechanisms associated with LN metastasis, and assisting in developing more effective therapeutic targets and biomarkers for detecting LN metastasis in patients with AM.

**Fig. 6 | Decoding the function of myeloid cells in AM. a** UMAP plot showing the landscape of myeloid cells of all tumor samples, color-coded by the annotated subclusters and split by LN⁺AM and LN⁻AM. **b** Feature plots illustrating classical marker genes used for the annotation of these subclusters. **c** Volcano plot displaying the DEGs in these annotated subclusters. Source data are provided in the Source Data file. **d** Pie plots showing the proportion of these subclusters of myeloid cells in the LN⁺AM and LN⁻AM groups. **e** GSVA analysis of selected hallmark pathways in these subclusters. Source data are provided in the Source Data file. **f** RNA velocity analysis demonstrating the evolutionary trajectory of these subclusters. **g** Pseudotime-ordered analysis of macrophages, colored by their subclusters. The

arrow indicates the potential evolutionary direction in the trajectory. **h** Heatmap displaying the expression levels of selected immunity/inflammatory genes in macrophage subclusters. **i** Feature plots showing the scores of M1/M2 polarization signatures in macrophage subclusters. **j** Box plots illustrating the scores of M1/M2 polarization signatures in macrophage subclusters. For M1/M2 score, the cell numbers for the *APOE*⁺Mac, *SPP1*⁺Mac, *CXCL10*⁺Mac and *F13A1*⁺Mac were 525, 263, 1028, 427, respectively. Significance was determined using a Kruskal-Wallis rank-sum test. The box center lines, bounds of the box, and whiskers indicate medians, first and third quartiles, and minimum and maximum values within 1.5× IQR of the box limits, respectively. Source data are provided in the Source Data file.

## Methods

### Patient specimens

For the scRNA-seq analysis, we included 12 patients diagnosed with AM who underwent curative surgical resection. Eight of these patients were randomly sourced from the Department of Plastic and Reconstructive Surgery at Zhongshan Hospital, Fudan University (FDZSH), and the Department of Musculoskeletal Oncology, Fudan University Shanghai Cancer Center (FDSCC). The remaining four patient datasets were obtained from the GEO database under accession number GSE189889[7]. The detailed clinicopathological characteristics of these patients are summarized in Supplementary Table 1. For ST-seq, five patients with AM undergoing curative surgical resection were randomly enrolled from the Department of Plastic & Reconstructive Surgery of FDZSH and the Department of Musculoskeletal Oncology of FDSCC. Detailed clinicopathologic characteristics of these patients were summarized in the Supplementary Table 2. Tumor cells in both primary and metastatic lesions were confirmed by pathologists via cytological detection during surgery and examination of paraffin sections after surgery. For LN⁻AM patients, we confirmed the absence of LN metastasis through multiple rigorous steps. First, we conducted thorough physical and imaging examinations on these patients before surgery. Second, during surgery, we obtained sentinel lymph nodes and performed comprehensive pathological examinations. Third, we maintained regular post-surgery follow-ups and reconfirmed the absence of lymph node metastasis through repeated physical and imaging examinations. All these meticulous procedures collectively affirm the absence of lymph node metastasis in LN⁻AM patients. None of these patients received targeted therapy, immunotherapy, or any other anti-tumor therapy prior to surgery. This study was conducted in accordance with the ethical standards of the Institutional Review Board of FDZSH and FDSCC. Written informed consents were obtained from all patients involved in this study for the use of their tissue samples and clinical information.

### Tissue microarray

A total of 138 paired melanoma and non-tumor tissues, along with an additional 58 melanoma tissues, including 101 AM tissues, were collected to construct the tissue microarray (TMA) as described in our previous study[55]. The tissues were histologically reviewed by two independent pathologists through H&E staining. and representative areas were pre-marked in the paraffin blocks.

### IHC

For IHC, tissue slices were deparaffinized and rehydrated using a gradient concentration of xylene and ethanol. After incubation with 0.3% hydrogen peroxide for 30 min, antigen retrieval was performed with citrate buffer at a sub-boiling temperature for 15 min, and then blocked with 5% bovine serum albumin for 60 min. The slices were incubated with primary antibody (anti-MITF, 1:2000 dilution, ab303530, ABCAM) overnight at 4 °C and incubated with HRP-conjugated secondary antibodies at 37 °C for 60 min. Detailed information of antibodies was provided in the Supplementary Table 3. Slices were then incubated with a 3, 3′-diaminobenzidine tetrahydrochloride (DAB) kit (Gene Tech, Shanghai, China) for color development and incubated with

hematoxylin for nuclear counterstaining. Images were obtained using the CaseViewer software (3DHISTECH, Budapest, Hungary).

### Multiplex IHC

For multiplex IHC, tissue slices were deparaffinized in xylene, rehydrated in a series of ethanol concentrations (100%, 90%, 70%), incubated with 0.3% hydrogen peroxide, antigen retrieved with citrate buffer, and blocked with 5% BSA. Then, the slices were incubated in a humidified chamber at 37 °C for 60 min with primary antibodies from three panels. Panel 1: anti-CD8 (1:300 dilution, BX50036, Biolynx), anti-CD4 (1:300 dilution, BX50023, Biolynx), anti-NCAM1 (1:100 dilution, 3576 S, CST), anti-FGFBP2 (1:1000 dilution, HPA039180, SIGMA); Panel 2: anti-FAP (1:50 dilution, BM5121, Boster), anti-CD31 (1:1000 dilution, 3528 S, CST); Panel 3: anti-Galectin-9 (1:300, NBP2-45619, NOVUS), anti-TIM3 (1:200, 45208 S, CST), anti-CD45 (1:200, BX50068, Biolynx), anti-CD44 (1:200, 3570 S, CST). Detailed information of antibodies was provided in the Supplementary Table 3. Slices were then incubated with the corresponding HRP-conjugated goat anti-mouse or goat anti-rabbit second antibodies (1:100 dilution, Leica Biosystems) at 37 °C for 10 min. Then the slide was again placed in citrate buffer to remove redundant antibodies before the next step. Finally, the slices were incubated with DAPI solution at 37 °C for 10 min in the dark. Images were captured using the CaseViewer software (3DHISTECH, Budapest, Hungary).

### Preparation of single-cell suspensions

Fresh primary lesions were isolated immediately following tumor resection and transferred to a 50 mL centrifuge tube filled with pre-cooled RPMI 1640 medium containing 0.04% bovine serum albumin (BSA, Gibco, Carlsbad, CA, USA). They were then quickly transported on ice to the FDZSH laboratory to minimize the ischemic time. Samples were cut into 1 mm³ pieces, followed by enzymatic digestion using the Miltenyi Tumor Dissociation Kit (Miltenyi, Bergisch Gladbach, Germany). The samples were then centrifuged at $300\,g$ for 30 s, and the supernatant was discarded. Next, 1× PBS (calcium and magnesium free) containing 0.04% BSA (400 µg/ml) was added, and centrifugation was performed at $300\times g$ for 5 min. The cell pellet was resuspended in 1 ml of red blood cell lysis buffer and incubated for 10 min at 4 °C. Subsequently, samples were resuspended in 1 ml of PBS containing 0.04% BSA and filtered using Scienceware Flowmi 40-µm cell strainers (VWR). Finally, 10 µl of suspension was examined under an inverted microscope for counting using a hemocytometer. Trypan blue was used to quantify the live cells.

### Droplet-based single-cell RNA sequencing

Cell suspension was subjected to Chromium Next GEM Single Cell 3′ Reagent Kit (version 3.1) for library preparation according to the manufacturer's protocol (10× Genomics, Pleasanton, CA). Single-cell libraries were sequenced on an Illumina Nova-Seq 6000 PE150 System (Illumina, San Diego, CA, USA) by paired-end sequencing. cDNAs were obtained after the GEM (Gel Bead-in-emulsion) generation and barcoding, followed by GEM reverse transcription (RT) reaction. Next, cDNA was amplified using polymerase chain reaction (PCR) for the appropriate cycles, depending on the number of

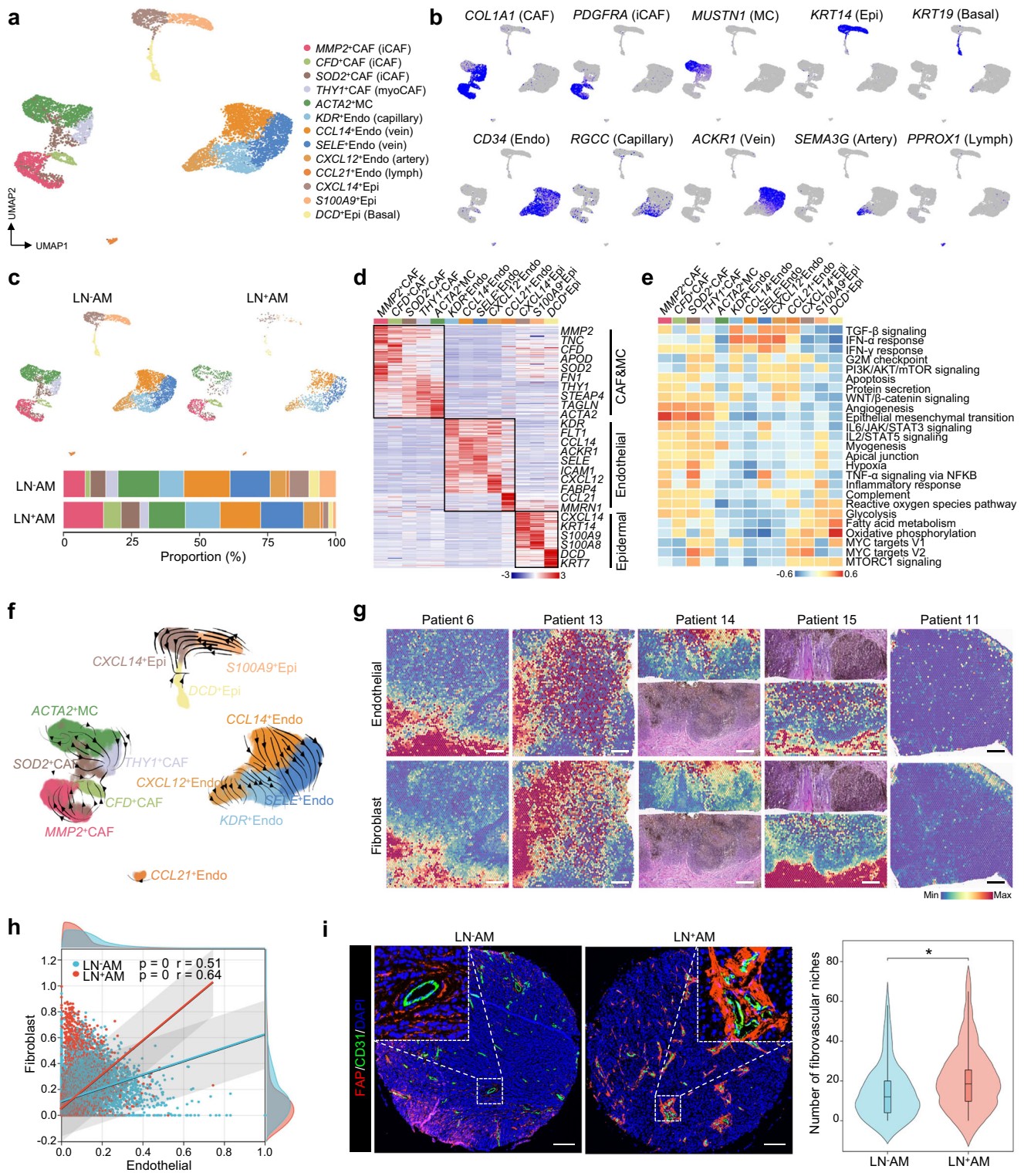

recovered cells. Subsequently, the amplified cDNA was fragmented, end-repaired, A-tailed, ligated to an index adaptor, and further amplified.

## scRNA-seq data processing

The Cell Ranger software pipeline (version 5.0.0) was used for demultiplexing, barcode processing, alignment, and initial clustering of the raw scRNA-seq profiles. Raw sequencing reads were mapped, annotated, and quantified using the GRCh38 reference annotation file (ENSEMBL, https://cf.10xgenomics.com/supp/cell-exp/

refdata-gex-GRCh38-2020-A.tar.gz). The unique molecular identifier (UMI) count matrix was processed using the R package Seurat[56] (version 3.1.1), and a unified standard was applied to filter cells with UMI/gene numbers out of the limit of the mean value +/− 2 folds of the median absolute deviation, assuming a Gaussian distribution of UMI/gene numbers of each cell. Following a visual inspection of the cell distribution, we further discarded low-quality cells, where more than a quarter of the counts belonged to mitochondrial genes. Additionally, the DoubletFinder package (version 2.0.2) was used to identify potential doublets[57]. To handle batch effects, we integrated

**Fig. 7 | Fibrovascular niches within the stromal compartment of AM. a** UMAP plot illustrating the landscape of all stromal cells color-coded by their associated subclusters. **b** Feature plots displaying classical marker genes used for the annotation of these subclusters. **c** UMAP plot showing the separation of stromal cells based on LN⁺AM and LN⁻AM, accompanied by a bar plot indicating the proportion of these subclusters in LN⁺AM and LN⁻AM. Source data are provided in the Source Data file. **d** Heatmap presenting the cluster-specific genes of these subclusters. Source data are provided in the Source Data file. **e** GSVA analysis of selected hallmark pathways in these subclusters. **f** RNA velocity analysis demonstrating the evolutionary trajectory of these subclusters. **g** Spatial feature plot showing the scores of endothelial cells and fibroblasts in tissue sections using ST-seq data. Scale bar, 1000 μm. **h** Scatter plot showing the correlation between endothelial (x-axis)

and fibroblast (y-axis) cells, stratified by LN⁺AM and LN⁻AM. The correlation is evaluated using the two-sided Spearman correlation coefficient. The gray band represents the 95% confidence interval of the regression line. Source data are provided in the Source Data file. **i** Representative image of fibrovascular niches detected by anti-FAP and anti-CD31 in AM sections using the multiplex IHC assay, and violin plot showing the number of fibrovascular niches in LN⁻AM (n = 73) and LN⁺AM (n = 28). The box center lines, bounds of the box, and whiskers indicate medians, first and third quartiles, and minimum and maximum values within 1.5× IQR of the box limits, respectively. Significance was determined using a two-sided Mann–Whitney U-test (*P-value < 0.05; P = 0.03). Scale bar, 150 μm. The experiment was repeated once with similar results. Source data are provided in the Source Data file.

the scRNA-seq data with the Canonical Correlation Analysis (CCA) algorithm[58]. After applying these quality control criteria, 52,382 high-quality single cells were used for downstream analysis. Normalized expression profiles of all samples were merged using the Merge() function in R (version 3.6.1). Subsequently, library size normalization was performed using the NormalizeData() function in Seurat to obtain a normalized count, and the results were log-transformed. The top 2000 highly variable genes were identified using the FindVariableFratures() function, according to a previously described method.

Top principal components (PCs) were computed based on the expression profiles of the top 2000 highly variable genes. The FindNeighbors() and FindClusters() functions in Seurat were used for cell clustering. RunUMAP() functions were used for visualization when appropriate. Cells were visualized using a 2D Uniform Manifold Approximation and Projection (UMAP) algorithm with the RunUMAP() and DimPlot() functions. Marker genes in each cluster were identified using the FindAllMarker() function in Seurat. For a given cluster, the FindAllMarkers() function identified positive markers compared with all remaining clusters.

### Single-cell copy-number variation evaluation
CNV of each cell on the chromosome was evaluated using the R package inferCNV (version 1.0.4)[39]. The CNV levels of these main cell types were calculated based on the amount of gene expression from the scRNA-seq data for each cell with a cut-off of 0.1. Genes were sorted based on their chromosomal locations, and the moving average of gene expression was calculated using a window size of 101 genes. The expression was centered at zero by subtracting the mean value. Melanoma cells were selected as the malignant cells, leaving all remaining cells as normal cells. The parameters of inferCNV analysis included "denoise," default hidden Markov model settings, and a value of 0.1 for "cutoff.".

### Trajectory and RNA velocity analysis
Developmental pseudo-time analysis was performed using the Monocle2 package (version 2.9.0)[60] to infer the developmental trajectory of the indicated cells. The raw count was first converted from the Seurat object to the CellDataSet object using the ImportCDS() function in Monocle. The DifferentialGeneTest() function was used to select ordered genes (q-value < 0.01) that were likely to be informative for the ordering of cells along the pseudotime trajectory. Dimensional reduction clustering analysis was performed using the reduceDimension() function, followed by trajectory inference with the orderCells() function using default parameters. Gene expression was plotted using the plot_genes_in_pseudotime() function to track the changes over time.

To recover the cellular dynamics of the indicated cells, we performed RNA velocity analysis using the Python script velocyto.py[61]. Spliced and un-spliced reads were counted using the Cell Ranger output folder. The calculation of RNA velocity values for each gene in

each cell and the embedding of the RNA velocity vector in low-dimensional space were performed using the R package velocyto.R (version 0.6). The velocity fields were projected onto the UMAP embedding obtained using Seurat.

### Simultaneous gene regulatory network analysis
SCENIC analysis was performed using the motif databases for RcisTarget and GRNBoost (SCENIC, version 1.1.2) with default parameters[62]. We identified transcription factor-binding motifs that were overrepresented on a gene list using the RcisTarget package. The activity of each regulon group for each cell type was scored using the AUCell package. To evaluate the cell type specificity of each predicted regulon, we calculated the regulon specificity score (RSS), which is based on the Jensen-Shannon divergence (JSD), a measure of the similarity between two probability distributions. Specifically, we calculated the JSD (Jensen-Shannon divergence) between each vector of binary regulon activity overlapping with the assignment of cells to a specific cell type. The connection specificity index (CSI) for all regulons was calculated using the scFunctions package.

### Cell-cell communication analysis with CellChat
Potential intercellular communication was assessed using the CellChat R package (version 1.1.3)[63]. The normalized expression matrix was imported to create a CellChat object using the CellChat() function. The data were then preprocessed with the identifyOverExpressedGenes(), identifyOverExpressedInteraction(), and ProjectData() functions using default parameters. The computeCommunProb(), filterCommunication(), and computeCommunProbPathway() functions were then used to determine any potential ligand-receptor interactions. Finally, the cell communication network was aggregated using the aggregateNet() function.

### Correlation to public dataset
The bulk RNA-seq data of 26 primary AM samples were retrieved from GSE162682 (www.ncbi.nlm.nih.gov/geo)[6]. For *MYC*⁺MEL and *FGFBP2*⁺NKT subgroups, genes with avg_log2FC > 0.5 and adj. P-value < 0.01 were considered as marker genes. Mean TPM levels of the marker genes were log2-transformed and used as gene signatures. Subsequently, relative cell abundance was divided into the top 50% and bottom 50%, corresponding to high and low abundance. Kaplan-Meier (KM) analysis was performed to evaluate the prognostic value of the cell clusters and determine the role of these cell clusters in melanoma progression. KM survival curves and P-values were obtained using the R package survival (version 3.5-5).

### Differential expression and pathway analysis
Differentially expressed genes (DEGs) were identified using the FindMarkers() function (test.use = presto) in the Seurat package[56]. To assign pathway activity estimates to individual cells, we applied GSVA using standard settings as implemented in the GSVA package (version 1.30.0)[64]. The GSEABase package (version 1.44.0) was used to load the

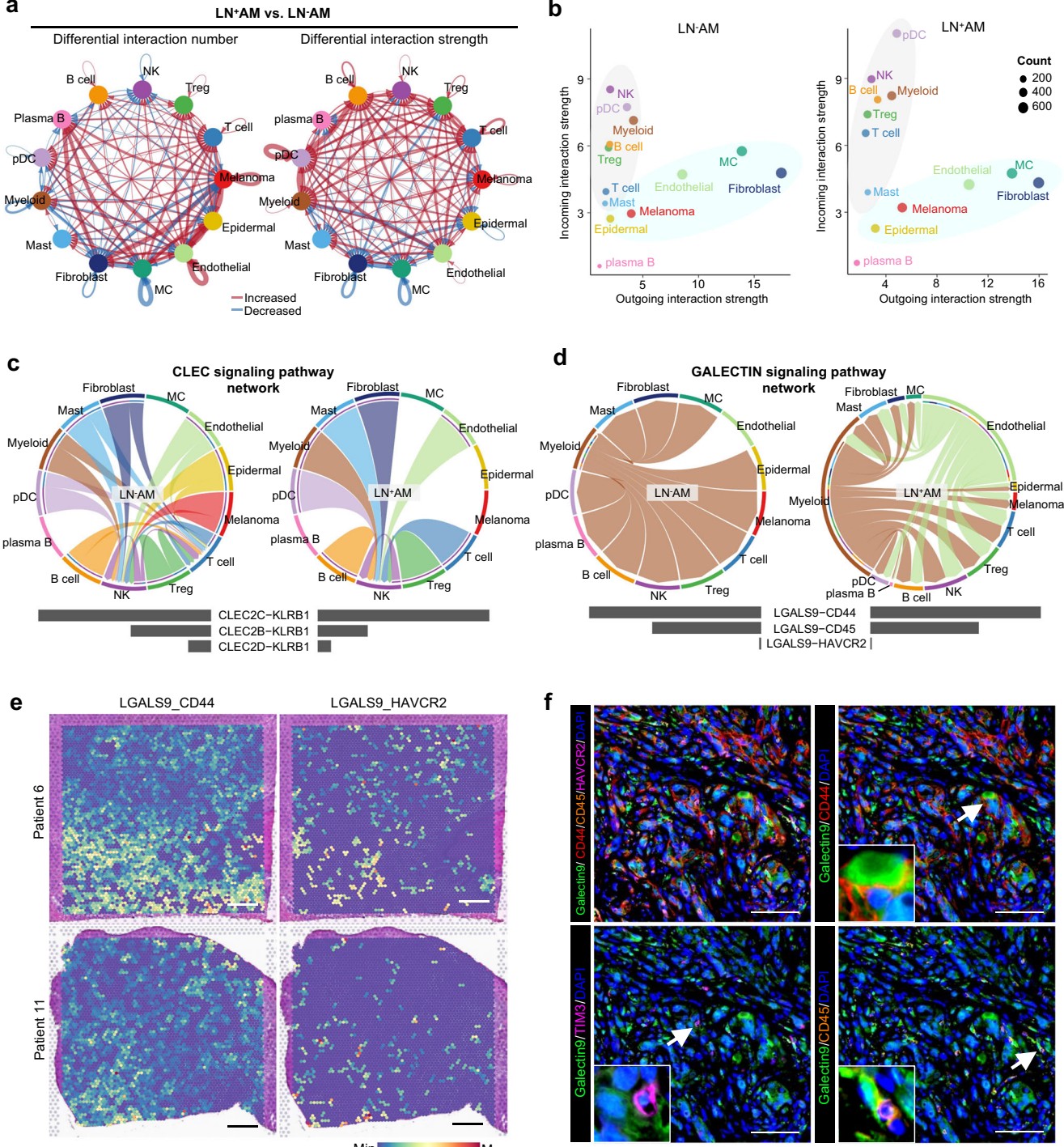

**Fig. 8 | Intercellular communication networks in AM. a** Comparison of the differential interaction number and strength of the 13 annotated cell types between LN⁺AM and LN⁻AM. The red line represents the increased interaction number or strength, and the bule line represents decreased interaction number or strength. Source data are provided in the Source Data file. **b** Scatter plots illustrating the incoming and outcoming interaction strength of the 13 annotated cell types in LN⁻AM and LN⁺AM. Source data are provided in the Source Data file. **c**, **d** Chord plots displaying the selected signaling pathway networks of CLEC (**c**) and GALECTIN (**d**) in LN⁺AM and LN⁻AM. Source data are provided in the Source Data file. **e** Spatial feature plots demonstrating the interaction activity of selected L-R pairs in tissue sections of patient 6 and patient 11 using ST-seq data. Scale bar, 1000 μm. **f** Representative images of L-R pairs (Galectin-9_CD44, Galectin-9_TIM-3 and Galectin-9_CD45) in AM cohort (n = 101) captured by the multiplex IHC assay. Galectin-9 is shown in green, CD44 in red, TIM-3 in purple, and CD45 in orange. The white arrow indicates a representative interaction of the selected L-R pairs. The experiment was repeated once with similar results. Scale bar, 100 μm.

gene set file, which was downloaded from the Kyoto Encyclopedia of Genes and Genomes database (https://www.kegg.jp/) and processed. Differences in the pathway activities scored per cell were calculated using the LIMMA package (version 3.38.3).

**Data analysis of ST-seq**
Tissue slice was printed with two identical capture areas from primary lesions of five patients with AM, including 3 LN⁺ and 2 LN⁻ patients. Tissue slices were permeabilized for 20 min, as defined by the tissue

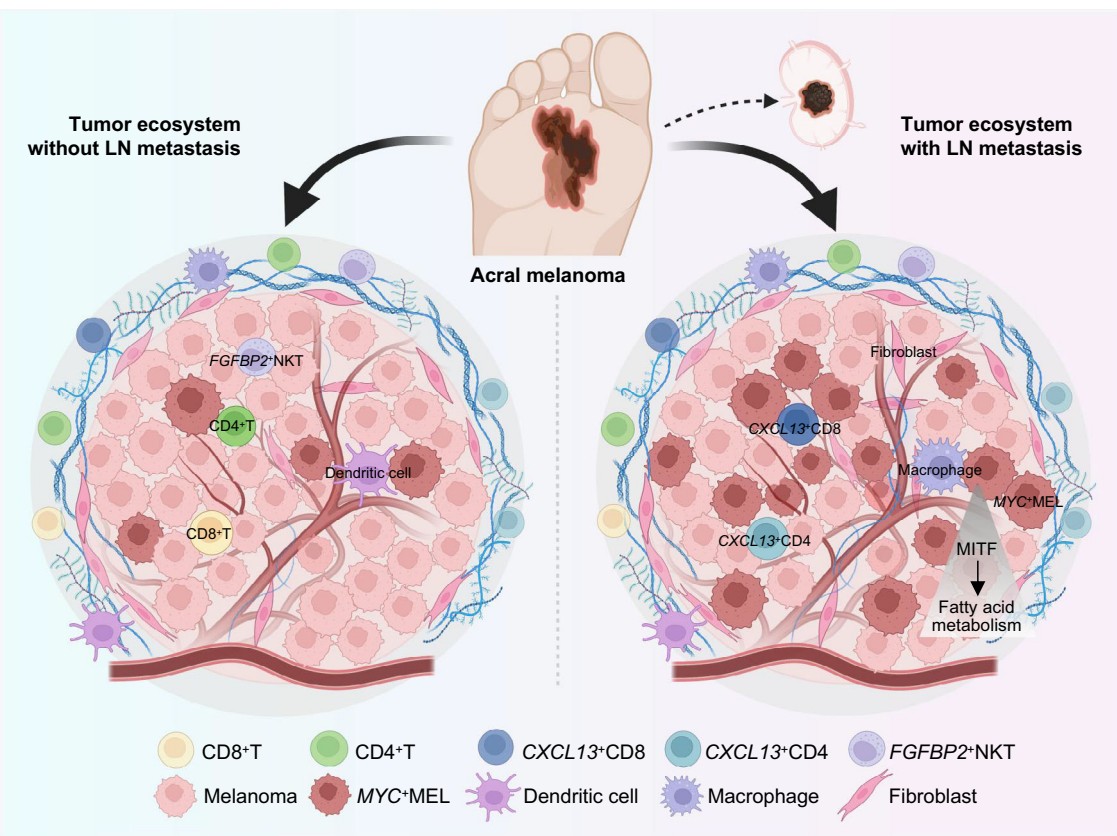

**Fig. 9 | Graphical abstract illustrating the distinct characteristics of the tumor ecosystem of AM with or without LN metastasis.** The primary tumor lesion of LN⁺AM exhibits a more suppressed TME, with decreased *FGFBP2*⁺NKT, CD4⁺T and CD8⁺T cells, and increased *CXCL13*⁺CD8⁺T and *CXCL13*⁺CD4⁺T cells, macrophages, and fibrovascular niches compared to that of LN⁻AM. *MYC*⁺*MEL* cells are significantly increased in the primary tumor of LN⁺AM, and MITF-mediated FAO activation drives tumor LN metastasis. A part of the image has been adapted from Biorender.com.

optimization flow performed in advance. Gene expression information for ST slices was captured using the Visium Spatial platform (10× Genomics) with spatially barcoded mRNA-binding oligonucleotides using the default protocol. Raw sequencing reads were checked for quality and mapped using Spaceranger (version 1.2.0). Normalization across spots was performed using the LogVMR() function. After quality control using Space Ranger software, the number of high-quality spots ranged from 4323 to 4992. At each spot, the average UMI number, gene number, and mitochondrial gene proportion ranged from 6354 to 19581, 1950 to 5642, and 2.43% to 6.96%, respectively. Dimensionality reduction and clustering were performed using independent component analysis (PCA) at a resolution of 1.1 with the first 30 PCs. Signature scoring derived from scRNA-seq or ST-seq signatures was performed using the AddModuleScore() function with the default parameters in Seurat. Spatial feature expression plots were generated using the spatial feature plot function in Seurat.

### Animal models

Male C57BL/6 mice aged 5 ~ 6 weeks ($n = 27$) were purchased from the SLAC laboratory animal company (Shanghai, China) and housed in a pathogen-free environment strictly following the 3 Rs guidelines (replacement, reduction, and refinement). Mice were housed in a controlled environment with a 12-h dark/light cycle and a temperature of 22 °C. They had ad libitum access to food and water, which were autoclaved. Cages were changed weekly. The humidity was monitored and maintained between 30% and 70%. All mice were randomized, and the investigators were blinded to the group assignment. All experimental procedures were approved by the Animal Experimentation Ethics Committee of FDZSH.

For the footpad implantation model, $5 \times 10^5$ B16F0-Vector or B16F0-*Mitf* cells were subcutaneously implanted into the footpad region of the hind limbs of C57BL/6 mice. The volume of foot pads (containing tumor lesions) was monitored and calculated using the formula: 0.5× (larger diameter) × (smaller diameter)². For FAO inhibition, Etomoxir (40 mg/kg, Sigma-Aldrich) in PBS (50 µl) was daily injected subcutaneously into the anterolateral side of the mouse leg, right above the hind limb foot where B16F0 cells were implanted. Treatment was initiated in the second week after tumor implantation. To increase the LN metastasis rate, particularly considering the lower LN metastasis rate after Etomoxir treatment, we made the decision to extend the study timeline to ensure an adequate LN positive rate for robust statistical analysis. Importantly, we still need to ensure that the maximum diameter of the tumor does not exceed 2 cm, and this has been approved by the Animal Experimentation Ethics Committee of FDZSH. On day 25, the mice were anesthetized with pentobarbital sodium (40 mg/kg), and the ipsilateral popliteal LNs and primary tumor tissues were obtained. Tissues were fixed in formalin and embedded in paraffin. Consecutive sections were prepared from each tumor tissue block and used for H&E and IHC experiments.

### Western blot analysis

For total protein extraction, cells were lysed in RIPA buffer supplemented with a protease inhibitor cocktail (Beyotime, P1005, 1:100). Total protein (10 µg) was loaded onto a Bis-Tris SDS/PAGE gel and transferred onto polyvinylidene difluoride (PVDF) membranes. The membranes were then blocked with 5% BSA for 1 h and incubated overnight at 4 °C with primary antibodies against MITF (ABCAM, ab303530, 1:1000) and Histone-H3 (Proteintech, 17168-1-AP, 1:8000).

Detailed information of antibodies was provided in the Supplementary Table 3. The membranes were then exposed to the secondary antibody (Beyotime, P0208, 1:1000) for 1 h. The bands were visualized using a DAB kit and analyzed with an imaging system.

## Statistical analysis

Statistical analyses were performed using R (version 4.1.1) and GraphPad Prism (version 9.0; San Diego, CA, USA). Unpaired student's $t$-test, Wilcoxon rank-sum test, Mann–Whitney $U$-test, one-way ANOVA test, Kruskal–Wallis rank-sum test, and Spearman correlation analysis were utilized for data analysis in this study. Cumulative survival time was estimated using the Kaplan-Meier estimator, and significance was assessed using the log-rank test. A two-sided test was used unless otherwise specified. A $P$-value of <0.05 was considered statistically significant.

## Reporting summary

Further information on research design is available in the Nature Portfolio Reporting Summary linked to this article.

## Data availability

The raw sequence data reported in this paper has been deposited in the Genome Sequence Archive (Genomics, Proteomics & Bioinformatics 2021) in National Genomics Data Center (Nucleic Acids Res 2022), China National Center for Bioinformation / Beijing Institute of Genomics, Chinese Academy of Sciences under the accession number HRA004456, which is publicly accessible. The raw sequencing data is available for non-commercial purposes under controlled access because of data privacy laws, and access can be obtained by request to the corresponding authors. For public datasets analysis, Li et al.'s dataset (including 4 AM samples) were retrieved from GSE189889. The bulk RNA-seq data of 26 primary AM samples were retrieved from GSE162682. Raw sequencing reads were mapped, annotated, and quantified using the GRCh38 reference annotation file [https://cf.10xgenomics.com/supp/cell-exp/refdata-gex-GRCh38-2020-A.tar.gz]. The remaining data are available within the Article, Supplementary Information or Source Data file. Source data are provided with this paper.

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

## Acknowledgements

The study was supported by project grants from the National Natural Science Foundation of China (82203528 to C.-Y.W.), the National Natural Science Foundation of China (82272891 to J.-Y.G.), the National Natural Science Foundation of China (81972559 to J.-Y.G.), the China Postdoctoral Science Foundation (2022M710769 and 2022TQ0072 to C.-Y.W.), Shanghai Sailing Program (22YF14 07400 to C.-Y.W.), Shanghai ShenKang Hospital Development Centre Project (SHDC2020CR2067B to J.-Y.G.). The authors thank OE Biotech Co., Ltd (Shanghai, China) for providing single-cell RNA-seq and spatial transcriptomics, and Dr. Xue Song and Xiao-hua Yao for assistance with bioinformatics analysis.

## Author contributions

J.-Y.G., Y.C., C.-Y.W., and F.-Z.Q. contributed to study design and supervised the study. C.-Y.W., W.S., and K.-J.S. contributed to writing the manuscript. C.-Y.W., K.-J.S., L.W., M.R., Y.-L.L., and Y.Z. assisted with data analysis. Y.-W.Y., Y.-H.Z., and W.-L.L. assisted in the preparation of the experiments. Y.X., T.H., Z.-X.G., and J.-Q.Z. aided in the collection of tissue samples. C.-Y.W., W.S., and K.-J.S. assisted with data collection. S.-L.Z., and M.Z. performed IHC and multiplex IHC staining and image analysis. R.-K.L., and Y.-Y.H. assisted in histopathological analysis.

## Competing interests

The authors declare no competing interests.
