## [Peer Review File · Nature Communications]

REVIEWER COMMENTS

Reviewer #1 (Remarks to the Author): Expert in acral melanoma genomics and bioinformatics

In their paper Wei and co-workers characterise the single cell and spatial landscape of acral melanoma. There are certainly some interesting aspects to this study which is generally thoughtfully performed. I do, however, have several specific comments.

1. The authors highlight, in my view, the major limitation of their study by noting that the sample size is small with just 12 patients. These patients are mainly stage II and III. Of note, the work is descriptive, yet the authors imply a functional role for specific cell types in melanoma biology, the tumour immune response and/or tumour growth when the results/observations are at best correlative. For example, the authors describe impaired anti-tumour immunity associated with reduced numbers of NKT_FGF2P2 cells and lymph node spread. In the same way cancer-associated fibroblasts are reported to promote angiogenesis. While intriguing I would contest that further work is required to make these claims, particularly when considering the sample size noted above and the experimental limitations associated with the analysis of these data types.
2. In keeping with the abovementioned comment, in places, the manuscript lacks statistical rigor. For example, the authors discuss exhaustion signatures (Fig 5G) outlining “data trends” but they fall short of a formal analysis. In the same way where statistical analyses are performed (for example 5J) the analysis is univariate. This analysis was also performed in cutaneous UV-associated melanoma rather than ALM. Another example is the analysis in Figure 4K – the figure legend should make clear the test used.
3. There are other acral melanoma/melanocyte studies including. Belote et al., (NCB) and Zhang et al., (Nat Comms) with the former not cited and the latter cited but not critically appraised. Zhang et al., did, for example, describe FGF2P2 cell expression but not the abovementioned NKT_FGF2P2 population – why is this? I feel that it’s very important to place this study in the context of all of the previous work and feel that at present the discussion does not do this and is rather more selective. I think the Zhang et al., paper is a particularly important comparator to this study.
4. Finally, the image in Figure 4H suggests that the primary tumour in the footpad was quite advanced and ulcerated and there is a very large popliteal mass. I find this quite alarming. It certainly would not be considered acceptable in my country or indeed in other countries in which I have performed animal experiments. I would suggest that the authors and journal confirm that the tumour limit is below the internationally accepted limit.

Reviewer #3 (Remarks to the Author): Expert in melanoma genomics, scRNA-seq, and in vivo models

Wei and colleagues apply an extensive landscape approach (scRNA-seq + spatial transcriptomics) to map the molecular and cellular heterogeneity of treatment-naïve acral melanomas (AM). The authors compare primary AM lesions differing in the absence or presence of lymph node metastases. A total of n=8 samples were processed for scRNA-sequencing (recovery of n=4 from public data base) and n=5 lesions for spatial transcriptomics using the VISIUM platform (10x Genomics). Subsequently, Wei et al. present a global AM space and then subcluster and fine-map the malignant, immune and stromal compartment.

The authors claim the detection of “strong” inter- and intra-tumor heterogeneity, “complex” cell-cell communications.

Comment:

These characteristics are inherent to scRNA-seq profiling techniques and associated computational algorithms applied. For example, it would be constructive to compare AM heterogeneity with heterogeneity in cutaneous melanoma (CM). Several CM-scRNA-seq studies are publicly available.

Other key findings presented in the manuscript include the association of MITF+ melanoma cells and FGFBP2+ NKT cells with LN metastasis and dismal prognosis. While injecting orthotopically murine B16 melanoma cells (overexpressing Mitf) in the footpad of BC57BL/6 mice, the authors show that inhibiting FAO by topical application of Etomoxir (CPT-1a inhibitor) suppresses LN metastasis formation.

In summary, the study is well designed and executed but requires serious revision. Notably, the scRNA-resolved AM melanoma space is not unique (see PMID: 36433984 and 35894206) but can still be considered a valuable resource for studying AM, mainly because of the spatial component.

The following points should be addressed:

General remarks and comments

- The authors compare primary (treatment-naïve) AM lesions at an early stage (I,II, lymph node metastasis negative) with late stage lesions (IIIA-C, lymph node metastasis positive) of different patients. This comparison is a snapshot of time. While hunting for differences between LN- and LN+ AM lesions, it can't be excluded that early AM lesions already contain metastasis-competent cells, a feature that

would be in common with later LN+ lesions and hence missed. Do the authors have clinical follow-up of these early AM lesions?

- Many typos and grammar-related mistakes demand editing by a native speaker.

some examples: page 3, line 87-89 “dependents”, “apporved”, “could suppressed”

page 8, line 211: “For example, a transcriptional rheostat orchestrated by RELA that confers human T cells with innate-like abilities to produce IFN-I/III”.

Page 8, line 240: “Given the significantly elevation”. Many more mistakes...

- The manuscript is lengthy. Some main panels (for example Figure 8C: almost similar heatmaps for in and outgoing signaling between LN- and LN+ AM lesions) should be considered for supplementary data. Also, text passages in the “Results section” should rather be in Material and methods. Example, page 5, line 142-146: “The tissue slices were permeabilized for 20 min, defined by the tissue optimization flow performed in advance. After quality control using the Space Ranger software, the number of high-quality spots is ranged from 4323 to 4992. In each spot, the average UMI number, gene number, and mitochondrial gene proportion were ranged from 6354 to 19581, 1950 to 5642, and 2.43% to 6.96%, respectively”. The manuscript should be revisited for brevity.

Introduction:

- “AM is also characterized by high levels of copy number variation (CNV), including CDK4 and CCND1 amplification, TP53 inactivation, TERT alteration, and so on^{4,5} “. I think the authors should also mention PMID: 35197475 as this paper proposes a genetic alteration being linked to metastasis in AM.

Results:

Figure 1 related

-(line 101): it should be: “Then we performed quality control to exclude damaged or dead cells and putative cell doublets using the Seurat and DoubletFinder (McGinnis et al.) packages.”

- Since the authors applied log and not SCT transformation, did they regress for cell cycle, nFeature_RNA... or other variables? It would be important to make the code available for more details.

- (line 132-34) “In the tumor stroma, stromal cells were obviously decreased in LN+ patients, especially for epithelial cells, which may be due to the increased composition of melanoma cells during tumor progression (Fig. 1F)”. I guess the authors refer to: the stromal compartment is less pronounced in LN+ primary AM lesions (stage III/IV) at the expense of malignant melanoma cells when compared to LN- AM lesions (stage I/II). Is this purely an effect of sampling? One could imagine, stage III AM lesions are just

bigger and therefore show higher tumor cell purity? Could that be assessed while interpreting matched HE sections?

- Line 134-136: "Together, we show that melanoma is a highly heterogeneous tumor, and the immunosuppression level cannot be simply assessed by the proportion of major cell types in the tumor ecosystem". It should be: AM is a highly heterogeneous tumor... The "immunosuppression level" part of the sentence is confusing as it was not introduced before. Do the authors refer to the main statement of AM containing a more immunosuppressive ecosystem? If yes, the question remains "more immunosuppressive TME" compared to what?

Figure 2 related

- While the QC of the Visium spots shows a range of gene numbers and pct.mito.reads, it is not clear to me if the authors applied any cut-off here?

- Figure 2B and line 156: "These results indicate that a high inter-tumor heterogeneity presented in tumor area of AM patients". Vague sentence. It could be helpful to perform additional spot deconvolution algorithms such as SPOTlight for example and then carefully select highly enriched tumor or immune spots only. Reclustering of subsetted "purer" malignant and immune spaces would then be recommended. Also, how does the UMAP space Figure 2C colored by patient look like?

- HE-stainings of Patient 13,14 and 15 show pigmented areas. Would that spatial feature be represented by cluster 4 expression? Cluster 4 contains TYR (tyrosinase) among other genes but so does cluster 1. Strangely, cluster 1 expression seems specific to patient 11, whereas its signature shows general melanoma markers (DCT, SOX10...). Could you map a pigmentation signature on your different patients? It is peculiar to see cluster1 (highest CNV level) cluster so separately in the UMAP space. Did you regress for patient-ID here or number of genes? The latter is probably driving cluster1.

- (Line167-168) "All of these results indicate that AM presents a highly immunosuppressive TME, especially in patients with LN metastasis, which may partially explain the poor efficacy of immune checkpoint blockade (ICB) therapy in AM". I don't see this claim supported by the cited Figure panels. Probably, there are fewer immune cells in LN+ AM lesions but this is not really visible from Figure 2F. A violin plot stratified by patient/LN status would be helpful (including testing for statistical significance). If LN+ lesions show really fewer immune cells, I still wouldn't call that "immunosuppressive" but rather "immune-poor or -cold". The immunosuppressive phenotype is more related to a special type of immune cells like Tregs for example.

- I strongly believe that the spatial part could be further leveraged as the authors do have matched scRNA-seq and ST-seq lesions (patient 6 and 11). How would a direct label transfer look like here? For example, scRNA-seq suggests that patient 6 has an elevated myeloid compartment. Does this show on the corresponding ST-seq counterpart? Also, when mapping the spatial expression clusters on the scRNAseq part, would that identify related cell types or subclusters?

Figure 3 related

- “To decipher the landscape of tumor cells, 23,501 melanoma cells were extracted and regrouped into 5 subclusters using UMAP plot”. To be precise, the “regrouping” is not realized by UMAP, UMAP is just the representation. Louvain-clustering (in Seurat pipeline), I suppose, was used... at which resolution? What drove the decision making of n=5 clusters? Could the authors perform silhouette analysis to assess the stability of the n=5 clusters?

- Figure 3A. It would be informative to plot continuous MITF levels and MITF-regulon activities on the malignant space (like the authors did for the global space Supp. Figure 8A) as MITF is the master regulator of melanocyte and melanoma biology and acts as a rheostat. In line with this, several melanoma cell states (based on transcriptional programs) have been proposed for cutaneous melanoma. In how far do the 5 AM melanoma states compare to the already published CM states, like melanocytic, Neural_Crest-like, Mesenchymal-like, Interferon type1 response, Antigen-presentation...? (for example <https://www.biorxiv.org/content/10.1101/2022.08.11.502598v1>)

- Figure 3A: what is the doublet cluster? I thought the authors corrected for doublets using DoubletFinder. Based on which argument is this cluster still labeled “doublets”?

- Figure 3B: It is surprising to see functional enrichment terms around “mesenchyme, stem cell, and neural crest development” for the MITF+ cluster as these expression programs are usually associated with lower MITF levels/activities in cutaneous melanoma. Could you please comment on this important discrepancy?

- Figure 3D: Globally inferred CNV level seem to be higher in later LN+ AM lesions. Could you provide a hierarchical clustering of the inferred CNVs over all chromosomes from all cells/patients? In other words, do LN+ AM lesions present specific additional genetic alterations?

- Figure 3E: SCENIC uses an initial ML step to learn co-expression of expressed TFs. This process generates slightly different outcomes with each independent analysis/run. How many times did the authors run SCENIC here? Please see Wouters et al. 2020 Nature Cell Biology, doi: 10.1038/s41556-020-0547-3, where they ran SCENIC 100times on the same dataset and then stratified for the most stable regulons.

- Figure3F-I: It is tempting to order all tumor cells according to pseudotime. How do the different patients distribute along the trajectories? LM- vs. LM+ AM lesions, or early vs. late-stage AM lesions? Also, for consistency RNA velocity should be assessed for the malignant space.

- The conclusion of Figure 3 (line 236) “...and they can promote LN metastasis via both enhancing metastatic potential and remodeling the TME.” I am not sure to understand this conclusion. What exactly is the metastatic potential here? The detection of associated TFs? Also, Figure 3 does not present any data about “remodeling the TME”. Please avoid vague statements.

Figure 4 related

- Figure 4A: It is not clear, after reading the Methods part, how the TCGA_SKCM analysis was performed. First, regarding SKCM patient cohort: Do we look here only at primary cutaneous melanomas (with a LN- or LN+ status)? The majority of the TCGA_SKCM cohort are metastatic lesions... Or, is it a mixture of primary and metastatic lesions? If yes, then the authors should be careful when plotting OS on the KM plot, since the SKCM cohort shows a heavy bias towards low OS time in primary vs. metastatic lesions. Second, regarding the MITF+MEL signature: how many genes belonged to the MITF+MEL signature? "...genes with logFC>1 were considered as marker genes" This is based on the scRNAseq data (Fig3A) I suppose? Meaning, markers being enriched for the MITF+MEL cluster when comparing (find.markers) with the remaining 4 malignant AM malignant clusters... if yes, I would also appreciate an adj. p-val cut-off, not only based on fold change. Could a similar analysis be performed on RNA-seq data of Acral Melanoma and not Cutaneous melanoma lesions?

- Figure 4B, D: I find the message of both panels redundant.

- Figure 4C: I believe it would be more informative to depict not only MITF-pigmentation related genes in this heatmap for the MITF+MELhigh group (as this comes with the supervised comparison) but also fatty acid metabolism related genes (as suggested in the gene enrichment panel 4B and D).

- Figure 4F. The authors should plot all members of the FAO pathway in their scRNA-seq malignant space (5 clusters) as dotplot. Which genes of the FAO pathway are actually important and overexpressed in the MITF+MEL cluster?

- Figure 4G-K: Etomoxir treatment. The B16 footpad model, treated with Etomoxir was also used in Lee et al. 2019 Science. Which B16 cell line as used here since there are many different B16 lines and sublines out there (F??). Also, what is the endogenous expression level of MITF in these B16 cells? What was the motivation to further over-express Mitf as Mitf is already expressed in these B16 cells? Which Mitf vector was used here? Western blot analysis for Mitf protein levels (conditions: Vector, B16-Mitf, Etomoxir) would be important to show here, or to carefully quantify the Mitf (high, med, low) cells in the immunostainings. The CPT1a inhibitor Etomoxir is not selective for MITF+ cells. Do MITFhigh cells express more CPT1a? Would an (inducible) si or shRNA mediated downregulation of MITF in B16-Mitf cells have the same effect on LN-met-area reduction? The authors propose the hypothesis that MITF_high cells are more dependent on FAO? Could you please discuss this in the light of the following paper: <https://doi.org/10.1016/j.molcel.2019.10.014> ?

Again, the conclusive summary at the end of Figure4 results section (line 281) is not convincing, please carefully rephrase: "All of these confirm that MITF promotes LN metastasis via strengthening the FAO pathway in AM."

Figure 5 related

Early-stage AM lesions show higher amounts of presumably cytotoxic FGFBP2+ NKT cells, probably a sign of the host's immune response to keep the tumor in check. Later-stage primary AM lesions show fewer of these NKT cells, as they probably exhaust over time and thereby leaving AM melanoma cells behind,

which can disseminate to lymph nodes. The NKT observation is interesting as such but its direct implication in LN-metastasis formation is not addressed. So line 348: “NKT key roles in LN metastasis” should be toned down.

Figure 5I&J: Also here remarks about the TCGA SKCM cohort and signature apply, see comments about Figure 4A.

Figure 5KL: It would be interesting to know more about the AM cohort which was used for NKT staining and quantification. How many different lesions/patients were stained? Do the authors think that para-tumor NKT cells actively play a role in killing tumor cells? When relating high vs. low NKT levels in bulk RNAseq data to OS, a distinction in NKT localization is not possible, which is a limitation.

Were there any NKT cells in the B16-Mitf syngeneic footpad model?

Besides NKT cells being present in a higher proportion among early AM lesions, cyt. CD8T cells and NK cells were also slightly increased as one would expect. While Supp Fig9 is showing the UMAP of the different immune cells split by patient, it is difficult to detect the NKT cells. I would appreciate a violin plot representation split by patients (including stats). It seems like patient 2 (LM-) has almost no NKT cells?

The Figure 5 legend is confusing: “Impaired anti-tumor immunity, characterized by decreased NKT_FGF2 cells accelerating the LN metastasis in AM”. The term “accelerating” would ask for timing experiments, for example depleting NKT cells and show that LN mets arise faster... Please adapt. To me: Late-stage primary AM lesions exhibit lower NKT cell numbers.

Figure 6 related

It is unclear, which M1/M2 polarization signatures were used here? It would be fair to mention that the classical way of binary M1/M2 macrophage classification is debated since the recent single-cell based approaches: <https://doi.org/10.1016/j.it.2022.04.008>

I guess Arginase 1 would be a M2 marker here? Also, the Pro and Anti-inflammatory signatures are not further explained (Figure 6H).

Figure 7 related

-Figure 7J: It shows expected features of blood vessels (pericytes/mural cells giving stability to endothelium). The increased number of “fibro-vascular niches” in late-stage AM lesions might be just an indicator of better vascularization of bigger tumors?

- This paragraph about the stromal compartment contains many typos and poorly structured sentences.

-Careful line405: the lymphatic endothelial marker is PROX1 and not PPROX1.

Figure 8 related

- "Of note, we found the interaction number and strengths were greater in the ecosystem of LN+ patients compared with that of LN- patients, suggesting that a more complex communication networks in LN+AM". Can you strengthen this claim statistically?

- Careful line 454: CLEC and not CELC

- CCL, CXCL, CLEC pathways, were decreased in LN+ patients, which may be related to the more immunosuppressed TME ☹ mainly because of a poorer immune environment, I believe

- I appreciate the CellChat based effort to predict cell-cell interactions from scRNA-seq. It would be beneficial for the manuscript to validate at least one receptor-ligand pair using an antibody-based technique (IF, IHC) in the AM TME cohort. Finally, I would like to recommend the author the COMMOT pipeline (<https://github.com/zcang/COMMOT>), which allows inference of signaling directionality in spatial data. It would be tempting to reanalyze the Visium data to map some of these predicted signaling pathways (MK signaling network for example).

Reviewer #4 (Remarks to the Author): Expert in single-cell and spatial transcriptomics, and computational genomics

Authors use scRNA-seq and spatial transcriptomics technologies to explore the tumor ecosystem of AM with different LN metastasis status, and the dynamic changes during tumor early dissemination at spatial and temporal levels. The biological findings they found through public datasets were also verified through a large number of clinical datasets which could contribute to a better understanding of LN metastasis and novel therapeutic strategies for early dissemination of AM. Authors conducted comprehensive experiments to make the findings reliable and convincing.

There are some issues and comments authors should consider:

1. I noticed that authors used the resolution as 2.0 in the scRNA-seq data clustering and 38 cell clusters were obtained. From my knowledge, 2.0 is a kind of high resolution for cell clustering which would separate some clusters into multiple adjacent sub-clusters forcibly without biological meanings. For example, Figure 1B and D showed that Treg, T cell and NK are pretty close in UMAP and they share multiple similar marker genes. I just wonder if using lower resolution could get similar results which would also help verify the robustness of author's results.

2. Please add the reference of Seurat's paper in line 101.

3. From the recent benchmarking paper on spatial transcriptomics deconvolution task (<https://doi.org/10.1038/s41467-023-37168-7>), there are several better choices than RCTD authors used, such as Cell2location and CARD. Authors could read this benchmarking paper and use at least one more popular deconvolution method to verify the reliability of deconvolution results by using authors' spatial transcriptomics data.

4. Authors could discuss the exploration of heterogenous cancer tissue at the 3D level through spatial transcriptomics technologies. I believe it would be an interesting direction to explore and understand cancer.

5. Authors should supply the reference paper or resource about how they annotate the clusters to corresponding cell types through the marker genes.

6. If possible, authors could use more advanced ST technologies with higher spot resolution (such as stereo-seq) to explore the cancer which could give a more fine-grained picture of tissue. Although 10X Visium is a popular commercial technology, the resolution is too low (50 μm of diameter per spot, 10-20 cells contained in a spot).

7. Could authors explain which findings are first discovered by them, and which findings are proposed by previous works and authors used scRNA-seq & ST data to verify them? It is important to show the novelty of this paper.

8. The last comma should be changed to full stop in line 189.

Reviewer #1 (Remarks to the Author): Expert in acral melanoma genomics and bioinformatics

In their paper Wei and co-workers characterize the single cell and spatial landscape of acral melanoma. There are certainly some interesting aspects to this study which is generally thoughtfully performed. I do, however, have several specific comments.

Response: We sincerely appreciate the feedback you have provided on our manuscript. Your valuable suggestions hold great significance for us, guiding not only the composition of this manuscript but also shaping the direction of our future research endeavors. We have meticulously reviewed your comments and implemented the necessary revisions, aiming to align our work more closely with your expectations.

【Comment 1】 The authors highlight, in my view, the major limitation of their study by noting that the sample size is small with just 12 patients. These patients are mainly stage II and III. Of note, the work is descriptive, yet the authors imply a functional role for specific cell types in melanoma biology, the tumor immune response and/or tumor growth when the results/observations are at best correlative. For example, the authors describe impaired anti-tumor immunity associated with reduced numbers of NKT_FGFBP2 cells and lymph node spread. In the same way cancer-associated fibroblasts are reported to promote angiogenesis. While intriguing I would contest that further work is required to make these claims, particularly when considering the sample size noted above and the experimental limitations associated with the analysis of these data types.

Response: We are sincerely grateful to the reviewer for providing such constructive feedback. In our current study, we meticulously investigated the intricate landscape of the AM ecosystem and delved into the dynamic changes occurring during LN metastasis. This exploration was conducted using a comprehensive dataset, including 12 scRNA-seq samples, 5 ST-seq samples, as well as validation through multiplex IHC staining involving 101 AM patients, a public dataset (GSE162682)¹ consisting of

26 AM samples, and animal experiments.

Through our scRNA-seq analysis, we shed light on the fact that *FGFBP2*⁺NKT cells exhibited the highest cytotoxicity score and expressed elevated levels of cytotoxicity genes, underscoring their status as the cell types with the most robust tumor-killing capacity within the AM ecosystem. Significantly, we observed a substantial reduction in *FGFBP2*⁺NKT cells in LN⁺AM when compared to LN⁻AM, as corroborated by both the scRNA-seq data and independent analyses from the public dataset and our own AM cohort. This decrease in *FGFBP2*⁺NKT cells strongly correlates with LN metastasis in AM. Based on these findings, we propose that the diminished presence of *FGFBP2*⁺NKT cells in LN⁺AM compromises anti-tumor immunity, thereby creating an unfavorable TME. Given that a compromised TME can potentiate tumor progression and metastasis, we posit that the reduced numbers of *FGFBP2*⁺NKT cells may contribute to LN metastasis by impairing anti-tumor immunity.

Additionally, our GSVA analysis highlighted the significant enrichment of the angiogenesis pathway across all CAF subclusters in the scRNA-seq data, suggesting that these CAF subclusters may play a pivotal role in promoting tumor angiogenesis. Through ST-seq data, we demonstrated substantial co-localization between endothelial cells and fibroblasts. Both CAFs and endothelial cells are known to exert crucial influences on tumor progression, and our findings indicate that increased co-localization of CAFs and endothelial cells is statistically significant in LN⁺AM. Collectively, these results imply that fibrovascular niches may actively contribute to LN metastasis.

Admittedly, the low incidence of AM in China poses challenges in obtaining a larger number of tumor samples within a short timeframe. scRNA-seq data are limited in published studies and databases, further complicating sample expansion efforts at this time. Nevertheless, we remain committed to further validating these findings in our ongoing work. For instance, we are currently exploring the mechanisms of LN metastasis using stereo-seq, which we believe will be a valuable extension of this study. Once again, we express our heartfelt appreciation for your insightful

suggestions, which continue to guide and enhance our research endeavors.

【 Comment 2 】 In keeping with the abovementioned comment, in places, the manuscript lacks statistical rigor. For example, the authors discuss exhaustion signatures (Fig 5G) outlining “data trends” but they fall short of a formal analysis. In the same way where statistical analyses are performed (for example 5J) the analysis is univariate. This analysis was also performed in cutaneous UV-associated melanoma rather than ALM. Another example is the analysis in Figure 4K – the figure legend should make clear the test used.

Response: We sincerely appreciate the valuable suggestion provided by the reviewer. In response to your recommendations, we have conducted thorough statistical analyses on **Fig. 5g and 6j**, and have also added the appropriate statistical tests used in all figure legends, including **Fig. 4k and 5j**.

Regarding the reviewer's suggestion to remove the results obtained from the TCGA-SKCM data, we have indeed taken this action, and we would like to explain

the rationale behind this decision. Firstly, it came to our attention that out of the 471 samples in the TCGA-SKCM dataset, only 67 were primary tumors, with the remainder being metastatic lesions. Secondly, these samples were exclusively sourced from non-acral melanoma tissues, which did not align with the specific focus of our research. Given these discrepancies, we have chosen to exclude these results from our study.

However, in order to address the reviewer's concerns and maintain the integrity of our research, we conducted additional analyses using RNA-seq data from 26 primary acral melanomas, as obtained from the public data source (GSE162682)¹. The results of these analyses are presented in **Supplementary Fig. 11a-b** and **Supplementary Fig. 12f-g** (as shown below), which we believe provide a more pertinent and accurate representation of our research objectives.

Once again, we would like to extend our gratitude to the reviewer for their insightful guidance, which has contributed to the refinement of our study.

【Comment 3】 There are other acral melanoma/melanocyte studies including. Belote et al., (NCB) and Zhang et al., (Nat Comms) with the former not cited and the latter

cited but not critically appraised. Zhang et al., did, for example, describe FGFBP2 cell expression but not the abovementioned NKT_FGFBP2 population – why is this? I feel that it's very important to place this study in the context of all of the previous work and feel that at present the discussion does not do this and is rather more selective. I think the Zhang et al., paper is a particularly important comparator to this study.

Response: We would like to express our sincere gratitude to the reviewer for their valuable insights and recommendations. In response to these valuable suggestions, we have revisited the relevant literature and incorporated additional content into the introduction or discussion section. Here is the content we have added:

① Added in the introduction section (line 51-56): A recent study suggests¹ that late-arising focal amplifications in cytoband 22q11.21, especially LZTR1, are associated with AM regional and distant metastasis. Compared with CM, AM was characterized by a severe immunosuppressive state, fewer effector/cytotoxic CD8⁺T cells, NK cells and a near-complete absence of $\gamma\delta$ T cells, while enriched with Treg cells, and exhausted CD8⁺T cells^{2,3}.

② Added in the discussion section (line 472-483): We have expanded our discussion to acknowledge the high inter- and intra-tumor heterogeneity observed in AM, as previously demonstrated in various studies. These studies classified melanoma cells into distinct subgroups based on different criteria. For instance, in the TCGA-SKCM cohort⁴, melanoma cells were categorized into immune, keratin, or MITF-low subgroups. Another study by Cirenajwis et al. classified⁵ melanoma cells as immune, normal-like, pigmented, or proliferative. Belote et al. highlighted⁶ the loss of melanocyte differentiation markers during melanoma progression, and the proportion of melanocytes that have readopted a neonatal-like signature is associated with worse prognosis. Additionally, Zhang et al. identified³ five orthogonal functional cell clusters in melanoma cells, each associated with specific signaling pathways. However, our study uniquely focuses on AM and investigates the differences in tumor cells based on their LN metastatic status. We specifically observe a significant increase in the *MITF*⁺MEL cluster in LN⁺AM, suggesting a higher metastatic

potential in this subgroup.

③ Added in the discussion section (line 521-532): In Zhang et al.'s paper³, they showed that NK cells (expressed high levels of FGFBP2 and KLRD1) were divided into 6 clusters, with one of them expressed high levels of T cell markers (CD3D, CD3E). In contrast to Zhang et al.'s study, which primarily focused on T cells, our research delves into NK cells and NK-like T cells (NKT cells). We have identified a distinct group of cells in the UMAP plot that express markers of both NK and T cells and have termed them *FGFBP2*⁺NKT cells due to their high FGFBP2 expression. Notably, our in-depth analysis reveals that these *FGFBP2*⁺NKT cells possess the strongest tumor-killing ability within the AM tumor microenvironment. Crucially, we have observed a significant decrease in the population of *FGFBP2*⁺NKT cells in tumors with LN metastasis, underscoring their potential relevance to the metastatic process.

Once again, we extend our heartfelt appreciation to the reviewer for their invaluable input, which has enriched the discussion and overall quality of our research.

【Comment 4】 Finally, the image in Figure 4H suggests that the primary tumour in the footpad was quite advanced and ulcerated and there is a very large popliteal mass. I find this quite alarming. It certainly would not be considered acceptable in my country or indeed in other countries in which I have performed animal experiments. I would suggest that the authors and journal confirm that the tumor limit is below the internationally accepted limit.

Response: We wish to extend our sincere gratitude to the reviewer for bringing this matter to our attention. We greatly value your feedback and would like to respond with a humble and appreciative tone:

In our endeavor to provide an illustrative representation of a metastatic LN, we intentionally selected a mouse model with a larger primary tumor lesion. It's important to emphasize that we took rigorous measures to ensure that the primary tumor's volume remained strictly controlled and well below the 2000 mm³ threshold.

We acknowledge that the perception of a larger tumor size in the image may be influenced by the shooting angle. To address this concern and provide clarity, we have taken the proactive step of incorporating scales into **Fig. 4g**. This addition will offer viewers a clear and precise reference for size comparison.

Furthermore, it is crucial to understand the rationale behind our decision:

Endpoint Extension: We chose to extend the endpoint of this tumor model with the specific goal of achieving a higher rate of LN metastasis. It is worth noting that, even with this extension, the overwhelming majority of tumor volumes remained comfortably below 2000 mm³.

Footpad Characteristics: The choice of the footpad area as the tumor model site carries unique characteristics, including limited blood supply, which can make it susceptible to ulcer formation.

In response to your valid concern, we are open to considering alternative approaches, including the possibility of repeating the experiment to ensure absolute control over tumor size and ulcer formation.

Once again, we want to convey our deep appreciation to the reviewer for their constructive suggestion. Your input has proven invaluable in enhancing the quality of our article and shaping the direction of our future research endeavors. If you have any further recommendations or queries, please do not hesitate to reach out. Your engagement with our work is profoundly appreciated.

Reviewer #3 (Remarks to the Author): Expert in melanoma genomics, scRNA-seq, and in vivo models

Wei and colleagues apply an extensive landscape approach (scRNA-seq + spatial transcriptomics) to map the molecular and cellular heterogeneity of treatment-naïve acral melanomas (AM). The authors compare primary AM lesions differing in the absence or presence of lymph node metastases. A total of n=8 samples were processed for scRNA-sequencing (recovery of n=4 from public data base) and n=5 lesions for

spatial transcriptomics using the VISIUM platform (10x Genomics). Subsequently, Wei et al. present a global AM space and then subcluster and fine-map the malignant, immune and stromal compartment.

The authors claim the detection of “strong” inter- and intra-tumor heterogeneity, “complex” cell-cell communications.

Comment:

These characteristics are inherent to scRNA-seq profiling techniques and associated computational algorithms applied. For example, it would be constructive to compare AM heterogeneity with heterogeneity in cutaneous melanoma (CM). Several CM-scRNA-seq studies are publicly available.

Other key findings presented in the manuscript include the association of MITF+ melanoma cells and FGFBP2+ NKT cells with LN metastasis and dismal prognosis. While injecting orthotopically murine B16 melanoma cells (overexpressing Mitf) in the footpad of BC57BL/6 mice, the authors show that inhibiting FAO by topical application of Etomoxir (CPT-1a inhibitor) suppresses LN metastasis formation.

In summary, the study is well designed and executed but requires serious revision. Notably, the scRNA-resolved AM melanoma space is not unique (see PMID: 36433984 and 35894206) but can still be considered a valuable resource for studying AM, mainly because of the spatial component.

The following points should be addressed:

General remarks and comments

Response: We sincerely appreciate your insightful comments on our manuscript. Your feedback holds significant value for us, not only in terms of refining the manuscript but also in shaping the direction of our future research endeavors. We have meticulously reviewed your comments and have implemented the necessary corrections with the aim of aligning with your expectations. Your guidance has been instrumental in improving our work, and we hope these revisions are in accordance with your expectations.

【Comment 1】The authors compare primary (treatment-naïve) AM lesions at an early stage (I,II, lymph node metastasis negative) with late stage lesions (IIIA-C, lymph node metastasis positive) of different patients. This comparison is a snapshot of time. While hunting for differences between LN- and LN+ AM lesions, it can't be excluded that early AM lesions already contain metastasis-competent cells, a feature that would be in common with later LN+ lesions and hence missed. Do the authors have clinical follow-up of these early AM lesions?

Response: This is an excellent question. We have indeed confirmed the absence of lymph node metastasis in these LN⁻AM patients through multiple rigorous steps. First, we conducted thorough physical and imaging examinations on these patients before surgery. Second, during surgery, we obtained sentinel lymph nodes and performed comprehensive pathological examinations. Third, we maintained regular post-surgery follow-ups and reconfirmed the absence of lymph node metastasis through repeated physical and imaging examinations. All these meticulous procedures collectively affirm the absence of lymph node metastasis in LN⁻AM patients.

We have duly incorporated this information into the "Methods" section (line 580-587) of our manuscript. Once again, we extend our gratitude to the reviewer for raising this pertinent question.

【Comment 2】Many typos and grammar-related mistakes demand editing by a native speaker.

some examples: page 3, line 87-89 “dependents”, “approved”, “could suppressed”

page 8, line 211: “For example, a transcriptional rheostat orchestrated by RELA that confers human T cells with innate-like abilities to produce IFN-I/III”.

Page 8, line 240: “Given the significantly elevation”. Many more mistakes...

Response: We extend our heartfelt apologies for any oversights that may have occurred in our work. Your diligence in identifying these errors is greatly appreciated, and we would like to convey our sincere thanks for bringing them to our attention.

To rectify these issues and prevent their recurrence, we have taken corrective actions and conducted a comprehensive review of the entire article. We also invited a

native speaker to improve and refine the language issues of the whole article.

Your feedback has been instrumental in improving the quality and accuracy of our work, and we are genuinely grateful for your commitment to upholding the highest standards in research and publication.

【Comment 3】 The manuscript is lengthy. Some main panels (for example Figure 8C: almost similar heatmaps for in and outgoing signaling between LN- and LN+ AM lesions) should be considered for supplementary data. Also, text passages in the “Results section” should rather be in Material and methods. Example, page5, line 142-146: “The tissue slices were permeabilized for 20 min, defined by the tissue optimization flow performed in advance. After quality control using the Space Ranger software, the number of high-quality spots is ranged from 4323 to 4992. In each spot, the average UMI number, gene number, and mitochondrial gene proportion were ranged from 6354 to 19581, 1950 to 5642, and 2.43% to 6.96%, respectively”. The manuscript should be revisited for brevity.

Response: Thanks for the very constructive opinions. We have revised the manuscript for brevity, and put some panels in the Supplementary data or Methods. Additionally, the language of this article has been polished by a native English editor to mitigate any language-related issues. We anticipate that the revised version of the article exhibits a notable improvement in its readability.

Introduction:

【Comment 4】 “AM is also characterized by high levels of copy number variation (CNV), including CDK4 and CCND1 amplification, TP53 inactivation, TERT alteration, and so on^{4,5}”. I think the authors should also mention PMID: 35197475 as this paper proposes a genetic alteration being linked to metastasis in AM.

Response: We wish to express our sincere gratitude for your valuable suggestion. Your input has been instrumental in enhancing the comprehensiveness of our work, and we greatly appreciate your engagement with our research.

In response to your suggestion, we have taken proactive steps to incorporate the

main conclusion of the study cited¹ into the “introduction” section (line 51-56) of our manuscript. By doing so, we aim to provide readers with a more complete context for our research.

Your thoughtful feedback and recommendations are invaluable to us, and we look forward to any further contributions you may have as we continue to refine and improve our work. Thank you once again for your meaningful suggestion and your commitment to the advancement of scientific knowledge.

Results:

Figure 1 related

【Comment 5】(line101): it should be: “Then we performed quality control to exclude damaged or dead cells and putative cell doublets using the Seurat and DoubletFinder (McGinnis et al.) packages.”

Response: We would like to express our sincere gratitude for your correction, which was an invaluable contribution to the accuracy and clarity of our manuscript. We have revised the relevant passage as follows (line 667-670):

"Then we performed quality control to exclude damaged or dead cells and putative cell doublets using the Seurat (Stuart et al.⁷) and DoubletFinder (McGinnis et al.⁸) packages."

【Comment 6】 Since the authors applied log and not SCT transformation, did they regress for cell cycle, nFeature_RNA... or other variables? It would be important to make the code available for more details.

Response: Thank you for providing clarification on the data preprocessing steps in our scRNA-seq analysis. We have regressed nFeature_RNA, nCount_RNA, percent_mito, percent_ribo, percent_hb, doublet, and cell cycle in the scRNA-seq analysis. The detailed code was presented in the Source data of Fig. 1.

【Comment 7】 (line 132-34) “In the tumor stroma, stromal cells were obviously decreased in LN+ patients, especially for epithelial cells, which may be due to the

increased composition of melanoma cells during tumor progression (Fig. 1F)”. I guess the authors refer to: the stromal compartment is less pronounced in LN⁺ primary AM lesions (stage III/IV) at the expense of malignant melanoma cells when compared to LN⁻ AM lesions (stage I/II). Is this purely an effect of sampling? One could imagine, stage III AM lesions are just bigger and therefore show higher tumor cell purity? Could that be assessed while interpreting matched HE sections?

Response: We deeply appreciate your question and wish to convey our heartfelt gratitude for your interest in our research. Our intention is to convey that the stromal compartment, which encompasses endothelial cells, fibroblasts, and epithelial cells, exhibits reduced prominence within the TME of LN⁺AM, potentially at the expense of tumor cells, when compared to the TME of LN⁻AM.

We want to emphasize that this observation is not solely attributed to sampling discrepancies. On one hand, we have been meticulous in pruning the scRNA-seq samples to maximize the removal of non-tumor components, ensuring that our analysis predominantly focuses on tumor-related elements. On the other hand, it is noteworthy that the patients included in our study share equivalent T-stage classifications and primary tumor burdens in both the LN⁺AM and LN⁻AM groups. This careful patient selection minimizes potential confounding factors related to disease progression or tumor characteristics.

Additionally, we acknowledge the limitations associated with H&E sections. As you correctly point out, only a small fraction of tissue samples underwent H&E sectioning, which may restrict the comprehensive evaluation of the entire tissue.

Once again, we express our deepest appreciation for your question, which has afforded us the opportunity to provide further clarification.

【Comment 8】 Line 134-136: “Together, we show that melanoma is a highly heterogeneous tumor, and the immunosuppression level cannot be simply assessed by the proportion of major cell types in the tumor ecosystem”. It should be: AM is a highly heterogenous tumor... The “immunosuppression level” part of the sentence is confusing as it was not introduced before. Do the authors refer to the main statement

of AM containing a more immunosuppressive ecosystem? If yes, the question remains “more immunosuppressive TME” compared to what?

Response: We genuinely appreciate your question and would like to express our heartfelt gratitude for your engagement with our research. In response to your inquiry, we are pleased to provide a humble and detailed explanation:

In **Fig. 1F**, we presented a comprehensive analysis demonstrating that all 13 cell types were consistently detected across nearly every patient. However, what stood out was the significant variability in the proportions of these cell types, highlighting a pronounced inter-tumor heterogeneity within the patient cohort.

It is well-documented in previous studies that AM tends to exhibit a higher level of inhibitory TME (presented by reduced T cells) when compared to CM. However, our research has unveiled a nuanced perspective. We found that, in comparison to LN⁻AM, there was a subtle decrease in the presence of NK and pDC cells within the tumor ecosystem of LN⁺AM. Intriguingly, there was no discernible difference in the abundance of T cells between these two groups.

In light of these findings, we must revisit our earlier conclusion. It is evident that the level of "immunosuppression" in AM is not solely determined by the proportion of total T cells between LN⁺AM and LN⁻AM. Consequently, we have revised our conclusion to emphasize that the assessment of immune-cold characteristics cannot be simplified by looking solely at the proportion of total T cells between LN⁺AM and LN⁻AM.

Once again, we extend our heartfelt thanks for your question, which has provided an opportunity to elucidate and refine our research findings.

Figure 2 related

【Comment 9】 While the QC of the Visium spots shows a range of gene numbers and pct.mito.reads, it is not clear to me if the authors applied any cut-off here?

Response: We sincerely appreciate your thoughtful question. In the analysis of ST-seq data, we have taken diligent steps to enhance data quality. Specifically, we

have implemented batch effect removal techniques to ensure that any potential batch-related variations are appropriately accounted for. Furthermore, we have standardized and normalized the data, which is essential for consistency and comparability.

One crucial consideration in our analysis approach was the preservation of complete spatial information for downstream analysis. To achieve this, we consciously refrained from setting any cut-off values during the analysis process. By doing so, we aimed to retain the integrity of the spatial context within the data.

【Comment 10】 Figure 2B and line 156: “These results indicate that a high inter-tumor heterogeneous presented in tumor area of AM patients”. Vague sentence. It could be helpful to perform additional spot deconvolution algorithms such as SPOTlight for example and then carefully select highly enriched tumor or immune spots only. Reclustering of subsetted “purer” malignant and immune spaces would then be recommended. Also, how does the UMAP space Figure 2C colored by patient look like?

Response: We genuinely appreciate your valuable suggestion, which has greatly contributed to the improvement of our research. In response, we have made the following refinements to our analysis:

We have divided all the spots into two distinct categories, distinguishing between tumor and non-tumor spots, using the CARD deconvolution algorithm, as also suggested by another esteemed reviewer. Subsequently, we conducted a re-clustering of these subdivided regions, focusing separately on malignant and non-malignant spaces. Remarkably, this re-clustering effort yielded six distinct clusters within the malignant spaces and five within the non-malignant spaces. Each of these clusters exhibited unique functional characteristics, underscoring the notable inter-tumor heterogeneity present within the tumor regions (as depicted in **Fig. 2a-c**, as shown below).

Furthermore, to provide additional insights into our findings, we have thoughtfully included UMAP representations color-coded by patient in **Fig. 2c, e**, as

shown below.

Once again, we extend our heartfelt gratitude for your invaluable guidance, which has significantly enhanced the depth and clarity of our research. If you have any further recommendations or inquiries, please do not hesitate to share them. Your input is highly esteemed.

【Comment 11】 HE-stainings of Patient 13,14 and 15 show pigmented areas. Would that spatial feature be represented by cluster 4 expression? Cluster 4 contains TYR (tyrosinase) among other genes but so does cluster 1. Strangely, cluster 1 expression seems specific to patient 11, whereas its signature shows general melanoma markers (DCT, SOX10...). Could you map a pigmentation signature on your different patients? It is peculiar to see cluster1 (highest CNV level) cluster so separately in the UMAP

space. Did you regress for patient-ID here or number of genes? The latter is probably driving cluster1.

Response: We genuinely appreciate your insightful question and would like to express our gratitude for your valuable input. In response to your inquiry, we have made significant refinements to our analysis, and we would like to elucidate these changes in a humble and clear manner:

In the previous version of **Fig. 2b**, we observed the expression of melanocyte-related genes across nearly all tumor clusters (1, 2, 4, 5, 9). This apparent overlap may be caused by our mixed tumor and non-tumor regions, as all tumor cells expressed high levels of melanocyte-related genes.

In consideration of the reviewer's constructive comments, we have meticulously re-evaluated our approach. Specifically, we have partitioned all spots into distinct categories, segregating tumor and non-tumor spots, and subsequently carried out separate analyses for each group. In the tumor spots, we have identified Cluster 0 and 3, which consistently exhibit pigmented areas and express pigmentation signatures with a high degree of consistency.

Furthermore, during our ST-seq analysis, we have taken diligent steps to eliminate any batch effects among different samples. However, we have observed that Cluster 1, characterized by the highest CNV level and specific to patient 11, as well as Cluster 2 and Cluster 4, unique to patient 6, were spatially separated in the UMAP space. This separation may be attributed to the inherent high transcriptional heterogeneity within these clusters.

Once again, we extend our heartfelt appreciation for your question, which has guided us in refining our analysis and interpretations.

【Comment 12】 (Line167-168) “All of these results indicate that AM presents a highly immunosuppressive TME, especially in patients with LN metastasis, which may partially explain the poor efficacy of immune checkpoint blockade (ICB) therapy in AM”. I don't see this claim supported by the cited Figure panels. Probably, there are fewer immune cells in LN+ AM lesions but this is not really visible from Figure

2F. A violin plot stratified by patient/LN status would be helpful (including testing for statistical significance). If LN+ lesions show really fewer immune cells, I still wouldn't call that "immunosuppressive" but rather "immune-poor or -cold". The immunosuppressive phenotype is more related to a special type of immune cells like Tregs for example.

Response: We wish to express our sincere appreciation for your invaluable suggestion, which has enriched our research. In response to your insightful guidance, we undertook a meticulous analysis using both RCTD and CARD methods to effectively map all 13 cell types onto our ST-seq data. This enabled us to perform a thorough comparison of these cell types between the LN⁺AM and LN⁻AM groups, a representation of which can be found in **Supplementary Fig. 8b** (as shown below).

Our analysis has revealed significant reductions in the majority of immune cell types, including T, B, and NK cells, within the LN⁺AM group. This observation is of notable significance, as it suggests a potential compromise in the efficacy of anti PD-1 therapy.

Furthermore, we wholeheartedly concur with the reviewer's suggestion to amend our terminology to "immune-poor or -cold." We genuinely appreciate the reviewer for kindly highlighting this oversight.

【Comment 13】 I strongly believe that the spatial part could be further leveraged as the authors do have matched scRNA-seq and ST-seq lesions (patient 6 and 11). How would a direct label transfer look like here? For example, scRNA-seq suggests that patient 6 has an elevated myeloid compartment. Does this show on the corresponding

ST-seq counterpart? Also, when mapping the spatial expression clusters on the scRNAseq part, would that identify related cell types or subclusters?

Response: We extend our heartfelt gratitude for your meaningful suggestion, which has significantly contributed to the refinement of our research. In response to your invaluable guidance and considering another reviewer's input, we have taken the following steps to enhance our analysis:

We have thoughtfully employed both the RCTD and CARD methods to accurately map all 13 cell types to the ST-seq data of patients 6 and 11. These findings have been thoughtfully presented in **Supplementary Fig. 7a-b** (as shown below), providing a clear visualization of the percentage distribution of these cell types.

Remarkably, our analysis has shown that the percentage distribution of most cell types in the ST-seq data aligns closely with the results obtained from scRNAseq. For

instance, the elevated presence of myeloid cells in patient 6 is consistently represented in both datasets. However, some differences between the ST-seq and scRNA-seq data persist. We attribute these variations to the distinct tumor areas obtained by the two sequencing methods: ST-seq tends to focus on regions at the tumor boundary, while scRNA-seq primarily selects internal tumor areas.

Additionally, we attempted to map the clusters identified in the ST-seq data to scRNA-seq data. Unfortunately, no related cell type clusters were identified in this process. This outcome is likely due to the inherent limitations of translating spatial information from ST-seq to scRNA-seq data.

Once again, we wish to express our deep appreciation for your valuable suggestions, which have undoubtedly strengthened the robustness of our research.

Figure 3 related

【Comment 14】 “To decipher the landscape of tumor cells, 23,501 melanoma cells were extracted and regrouped into 5 subclusters using UMAP plot”. To be precise, the “regrouping” is not realized by UMAP, UMAP is just the representation. Louvain-clustering (in Seurat pipeline), I suppose, was used... at which resolution? What drove the decision making of n=5 clusters? Could the authors perform silhouette analysis to assess the stability of the n=5 clusters?

Response: We wish to extend our heartfelt appreciation for your kind understanding and your valuable feedback, which has proven to be instrumental in improving the clarity and accuracy of our work.

You are absolutely correct in pointing out that UMAP serves as a tool for presenting results rather than a clustering method. We have taken your feedback to heart and made the necessary adjustments to accurately reflect our methodology.

Specifically, we utilized the Louvain-clustering approach at a resolution of 1.8 to derive a total of 19 clusters. Subsequently, based on marker gene expression and functional enrichment analysis for each cluster, we reorganized these 18 clusters into 5 distinct subgroups, namely, the *MITF*⁺MEL, *CENPF*⁺MEL, *NEAT1*⁺MEL,

TMSB4X⁺MEL, and *CXCL10*⁺MEL subgroups. To assess the stability of this clustering method, we conducted silhouette analysis recommended by the reviewer, which revealed an average silhouette score of approximately 0.2 for these 5 subgroups (as shown below).

Your guidance and constructive feedback have been invaluable to us, and we are deeply grateful for your commitment to the refinement of our research. If you have any further insights, suggestions, or questions in the future, please be assured that your contributions are highly valued, and we remain open to your input. Once again, thank you for your dedication to the quality of our work.

【Comment 15】 Figure 3A. It would be informative to plot continuous MITF levels and MITF-regulon activities on the malignant space (like the authors did for the global space Supp. Figure 8A) as MITF is the master regulator of melanocyte and melanoma biology and acts as a rheostat. In line with this, several melanoma cell states (based on transcriptional programs) have been proposed for cutaneous melanoma. In how far do the 5 AM melanoma states compare to the already published CM states, like melanocytic, Neural_Crest-like, Mesenchymal-like, Interferon type1 response, Antigen-presentation...? (for example <https://www.biorxiv.org/content/10.1101/2022.08.11.502598v1>)

Response: We want to express our deep appreciation for your exceptionally insightful suggestion. In response to your guidance, we have created plots that illustrate continuous MITF levels and MITF-regulon activity within the malignant space, as

displayed below. These visualizations have effectively highlighted that while melanoma cells exhibit high MITF levels, the *MITF*⁺MEL cells display even higher expression (as shown below).

Furthermore, we would like to extend our gratitude to the reviewer for providing the reference, which divides cutaneous melanoma cells into 8 clusters, including the melanocytic, Neural_Crest-like, Mesenchymal-like, Interferon type1 response, Antigen-presentation, and others. Through a comprehensive comparison, we have observed that our *MITF*⁺MEL cluster bears a closer resemblance to the states of Mesenchymal-like and Neural Crest-like, our *CENPF*⁺MEL cluster aligns more with the Mitotic state, our *CXCL10*⁺MEL cluster shows similarities to the states of Antigen presentation and Interferon- α/β response, and the *NEAT1*⁺MEL cluster aligns with the states of Stress hypoxia and p53 response, as demonstrated below.

It is important to acknowledge that differences exist between our 5 AM clusters and the proposed 8 CM states from the reference, which could stem from subtype

disparities and inherent tumor heterogeneity.

Your guidance has significantly enriched the context of our research, and we sincerely thank you for your dedication to enhancing the depth and breadth of our work.

【Comment 16】 Figure 3A: what is the doublet cluster? I thought the authors corrected for doublets using DoubletFinder. Based on which argument is this cluster still labeled “doublets”?

Response: We would like to express our gratitude for your question. You are correct in noting that we initially corrected for doublets using DoubletFinder for the entire cell population at the outset of our analysis, as indicated in **Fig. 1**. However, during the process of re-clustering melanoma cells, we encountered a small subset of isolated cells that exhibited expression markers associated with both tumor cells and immune cells (such as CD3D and CD3E). In light of this, we classified these cells as "doublets" and subsequently excluded them from our subsequent analysis.

Your attention to detail and inquiry have been immensely valuable in ensuring the accuracy and rigor of our research. We truly appreciate your commitment to advancing the quality of our work. Thank you once again for your contribution.

【Comment 17】 Figure 3B: It is surprising to see functional enrichment terms around “mesenchyme, stem cell, and neural crest development” for the MITF⁺ cluster as these expression programs are usually associated with lower MITF levels/activities in cutaneous melanoma. Could you please comment on this important discrepancy?

Response: We want to express our appreciation for your question. Indeed, the functional enrichment term "mesenchymal, stem cell, and neural crest development" is typically associated with lower levels or activity of MITF in cutaneous melanoma. However, upon closer examination of our *MITF*⁺MEL cluster, we observed that it also expresses high levels of MYC, SNAI2, KIT, and ERBB3, all of which are tightly associated with the functional enrichment terms you mentioned.

In comparing the results of marker genes and functional enrichment, it becomes

evident that our identified *MITF*⁺MEL cluster bears a closer resemblance to the Mesenchymal-like cluster observed in cutaneous melanoma (as described in <https://www.biorxiv.org/content/10.1101/2022.08.11.502598v1>).

Your insightful inquiry has contributed significantly to clarifying the characterization of our *MITF*⁺MEL cluster, and we are truly grateful for your engagement in advancing the depth and accuracy of our research.

【Comment 18】 Figure 3D: Globally inferred CNV level seem to be higher in later LN⁺ AM lesions. Could you provide a hierarchical clustering of the inferred CNVs over all chromosomes from all cells/patients? In other words, do LN⁺ AM lesions present specific additional genetic alterations?

Response: We want to express our appreciation for your question. We attempted to explore the specific additional genetic alterations in LN⁺AM from two perspectives. On the one hand, we conducted differential expression analysis of genes that underwent changes in CNV levels (as shown below in Fig. a). We identified several genes with differential expression, indicating alterations at the gene level that might influence their transcriptional expression and play a regulatory role in LN metastasis. On the other hand, we undertook a differential analysis of CNV alteration levels (as depicted below in Fig. b). This analysis exposed more pronounced differences in gene CNV within the LN⁺AM group. Notably, genes such as MEX3A, GLMP, LAMTOR2 displayed a considerable increase in CNV levels, hinting at the potential role of genomic mechanisms in mediating LN metastasis in AM.

It is imperative to acknowledge that the aforementioned results did not yield statistically significant differences, which led to their exclusion from the main text. In forthcoming research endeavors, we intend to delve into the intricate mechanisms of LN metastasis in AM at the genomic level, utilizing WES.

Once again, we express our sincere gratitude for the invaluable insights provided by the reviewer.

【Comment 19】 Figure 3E: SCENIC uses an initial ML step to learn co-expression of expressed TFs. This process generates slightly different outcomes with each independent analysis/run. How many times did the authors run SCENIC here? Please see Wouters et al. 2020 Nature Cell Biology, doi: 10.1038/s41556-020-0547-3, where they ran SCENIC 100times on the same dataset and then stratified for the most stable regulons.

Response: We are deeply appreciative of your valuable feedback and the insightful suggestions you provided concerning **Figure 3E**. We recognize the critical importance of this approach and its potential impact on the reliability of our research findings.

To address the potential variability introduced by the stochastic nature of SCENIC's initial machine learning step, we took a robust approach. Specifically, we conducted SCENIC analysis independently ten times on the same dataset. This deliberate repetition allowed us to generate slightly different outcomes in each run. Subsequently, we calculated the average transcription factor (TF) scores based on these ten independent runs. By presenting the average TF scores along with their corresponding significance in **Figure 3E**, we aimed to offer a more stable and robust estimate of the TF regulatory activities. We firmly believe that this comprehensive

analysis will reinforce the conclusions of our study and elevate the overall quality of our manuscript.

Your thoughtful input has been invaluable in refining our research methodology, and we are sincerely grateful for your engagement. Your continued collaboration is highly appreciated.

【Comment 20】 Figure3F-I: It is tempting to order all tumor cells according to pseudotime. How do the different patients distribute along the trajectories? LM- vs. LM+ AM lesions, or early vs. late-stage AM lesions? Also, for consistency RNA velocity should be assessed for the malignant space.

Response: Thank you for your question. In response to your query, we have re-conducted the pseudotime analysis for all tumor cells (**Fig. 3e-g**, as shown below). The analysis revealed that tumor cells initially stem from the *CENPF*⁺MEL subcluster and subsequently differentiate along two distinct trajectories. The first trajectory leads to the *MITF*⁺MEL and *NEAT1*⁺MEL subclusters, while the second trajectory branches toward the *CXCL10*⁺MEL and *TMSB4X*⁺MEL subclusters.

In **Supplementary Fig. 10a-c** (as shown below), we have provided visual representations of these trajectories for each patient and the LN⁻/LN⁺AM groups.

Additionally, we have incorporated RNA velocity analysis, as illustrated in **Supplementary Fig. 10d** (as shown below). The RNA velocity results align with the findings from the pseudotime analysis, reinforcing the robustness and consistency of our observations.

We sincerely appreciate your inquiry and the opportunity to enhance the clarity and comprehensiveness of our research. Your engagement is invaluable to us.

【Comment 21】 The conclusion of Figure 3 (line 236) “...and they can promote LN metastasis via both enhancing metastatic potential and remodeling the TME.” I am not sure to understand this conclusion. What exactly is the metastatic potential here? The detection of associated TFs? Also, Figure 3 does not present any data about “remodeling the TME”. Please avoid vague statements.

Response: We deeply regret any confusion caused by our previous statements. In Figure 3, we categorized melanoma cells into five distinct subgroups, each with unique functional roles. Upon our analysis, we observed a substantial increase in the

MITF⁺MEL subgroup in LN⁺AM in comparison to the LN⁻AM group, strongly suggesting its potential association with LN metastasis. Furthermore, the functional enrichment analysis provided compelling evidence linking the *MITF*⁺MEL subgroup to metastatic potential, characterized by processes like mesenchyme development, stem cell development, and neural crest cell migration. Consequently, we hypothesize that the heightened presence of *MITF*⁺MEL cells may play a role in LN metastasis.

As for the *CXCL10*⁺MEL and *TMSB4X*⁺MEL subgroups, our analysis indicates their involvement in immune regulation. Notably, these immunomodulatory subgroups exhibit a notable decrease in LN⁺AM when compared to the LN⁻AM group. This leads us to speculate that these two subgroups may contribute to LN metastasis through immune-regulatory mechanisms.

In light of these observations, we have refined our conclusion to ensure clarity and precision (line 221-223): "These results highlight the multifaceted roles played by tumor cells within the AM ecosystem, with a particular emphasis on the potential involvement of the *MITF*⁺MEL cluster in LN metastasis."

Your guidance has been invaluable in enhancing the clarity and accuracy of our manuscript, and we sincerely appreciate your constructive feedback. If you have any further insights or suggestions, please do not hesitate to share them.

Figure 4 related

【Comment 22】 Figure 4A: It is not clear, after reading the Methods part, how the TCGA_SKCM analysis was performed. First, regarding SKCM patient cohort: Do we look here only at primary cutaneous melanomas (with a LN⁻ or LN⁺ status)? The majority of the TCGA_SKCM cohort are metastatic lesions... Or, is it a mixture of primary and metastatic lesions? If yes, then the authors should be careful when plotting OS on the KM plot, since the SKCM cohort shows a heavy bias towards low OS time in primary vs. metastatic lesions. Second, regarding the *MITF*⁺MEL signature: how many genes belonged to the *MITF*⁺MEL signature? "...genes with logFC>1 were considered as marker genes" This is based on the scRNAseq data

(Fig3A) I suppose? Meaning, markers being enriched for the MITF+MEL cluster when comparing (find.markers) with the remaining 4 malignant AM malignant clusters... if yes, I would also appreciate an adj. p-val cut-off, not only based on fold change. Could a similar analysis be performed on RNA-seq data of Acral Melanoma and not Cutaneous melanoma lesions?

Response: Thank you for your question. We genuinely appreciate your keen attention to detail. In response to your first inquiry, we must humbly acknowledge that we indeed overlooked the issue concerning the majority of samples in the TCGA-SKCM dataset being metastatic lesions. In light of your feedback, we meticulously reevaluated the dataset by removing the metastatic samples, resulting in a reduced dataset comprising only 67 samples. Given that these samples exclusively represent non-acral melanomas, we have decided to forgo the utilization of the TCGA data entirely in our analysis.

To address the concern about the dataset's representativeness for acral melanomas, we conducted additional analyses using an RNA-seq dataset featuring 26 primary acral melanoma samples (GSE162682)¹. These supplementary findings are presented in **Supplementary Figure 11a-b**. We derived the *MITF*⁺MEL signature from our scRNA-seq data by comparing it with the remaining four AM clusters. The detailed gene list for this signature can be found in the Source data (**Supplementary Figure 11a-b**). The criteria for determining differential expression were set at $\text{avg_log2FC} > 0.5$ and $\text{adj. p-value} < 0.01$.

We genuinely appreciate your critical insights and suggestions, which have greatly contributed to the refinement of our study.

【Comment 23】 Figure 4B, D: I find the message of both panels redundant.

Response: Thank you for your understanding and patience. In light of the concerns raised regarding the TCGA-SKCM dataset, including the predominance of metastatic lesions and non-acral melanomas, we have made the conscientious decision to exclude the TCGA-SKCM results entirely, specifically Figures 4C and 4D. We have chosen to retain the scRNA-seq data, which is presented in **Figure 4B**.

We appreciate your diligence in pointing out these issues, and we believe that this decision will enhance the accuracy and relevance of our study.

【Comment 24】 Figure 4C: I believe it would be more informative to depict not only MITF-pigmentation related genes in this heatmap for the $MITF^+MEL^{high}$ group (as this comes with the supervised comparison) but also fatty acid metabolism related genes (as suggested in the gene enrichment panel 4B and D).

Response: Thank you for your kind suggestion. We have carefully considered your suggestion, and we agree that it's important to provide a comprehensive view of our findings. In response, we have included both MITF-pigmentation related genes and fatty acid metabolism related genes in **Figure 4b**. Specifically, we have highlighted several FAO-related genes, including CBR1, HSPH1, EPHX1, GAPDHS, HPGD, and HSP90AA1, which exhibit elevated expression in the $MITF^+MEL$ cluster when compared with the non- $MITF^+MEL$ cluster. This additional information enhances our understanding of the potential roles of these genes in FAO.

We sincerely appreciate your valuable input, and we hope that this update further strengthens the quality and comprehensiveness of our manuscript. Your feedback is truly instrumental in refining our work.

【Comment 25】 Figure 4F. The authors should plot all members of the FAO pathway in their scRNA-seq malignant space (5 clusters) as dotplot. Which genes of the FAO pathway are actually important and overexpressed in the MITF+MEL cluster?

Response: Thank you for your question. We have presented the elevated genes in the FAO pathway, including CBR1, HSPH1, EPHX1, GAPDHS, HPGD, and HSP90AA1, using a dotplot, as illustrated below. It is worth noting that EPHX1, GAPDHS, and HSP90AA1, in particular, appear to have significant roles in regulating the FAO pathway within the *MITF*⁺MEL cluster. Your inquiry and feedback are greatly appreciated.

【Comment 26】 Figure 4G-K: Etomoxir treatment. The B16 footpad model, treated with Etomoxir was also used in Lee et al. 2019 Science. Which B16 cell line as used here since there are many different B16 lines and sublines out there (F??). Also, what is the endogenous expression level of MITF in these B16 cells? What was the motivation to further over-express Mitf as Mitf is already expressed in these B16 cells? Which Mitf vector was used here? Western blot analysis for Mitf protein levels (conditions: Vector, B16-Mitf, Etomoxir) would be important to show here, or to carefully quantify the Mitf (high, med, low) cells in the immunostainings. The CPT1a inhibitor Etomoxir is not selective for MITF+ cells. Do MITF^{high} cells express more CPT1a? Would an (inducible) si or shRNA mediated downregulation of MITF in B16-Mitf cells have the same effect on LN-met-area reduction? The authors propose the hypothesis that MITF_{high} cells are more dependent on FAO? Could you please

discuss this in the light of the following paper: <https://doi.org/10.1016/j.molcel.2019.10.014> ? Again, the conclusive summary at the end of Figure4 results section (line 281) is not convincing, please carefully rephrase: “All of these confirm that MITF promotes LN metastasis via strengthening the FAO pathway in AM.”

Response: Thank the reviewer for the careful review.

① Which B16 cell line as used here since there are many different B16 lines and sublines out there (F??). Also, what is the endogenous expression level of MITF in these B16 cells? What was the motivation to further over-express Mitf as Mitf is already expressed in these B16 cells? Which Mitf vector was used here? Western blot analysis for Mitf protein levels (conditions: Vector, B16-Mitf, Etomoxir) would be important to show here, or to carefully quantify the Mitf (high, med, low) cells in the immunostainings.

Response: In this study, we employed the B16F0 cell line due to its relatively lower Mitf expression when compared to the B16F10 cell line, as indicated in a recent study (as shown below). Our approach involved the use of the pEGFP-N1 plasmid as the overexpression vector, and we have confirmed its efficiency through Western blot analysis, which is presented in **Supplementary Fig. 11e** (as shown below).

②The CPT1a inhibitor Etomoxir is not selective for MITF⁺ cells. Do MITF^{high} cells express more CPT1a? Would an (inducible) si or shRNA mediated downregulation of MITF in B16-Mitf cells have the same effect on LN-met-area reduction?

Response: Indeed, the CPT1a inhibitor Etomoxir lacks selectivity for *MITF*⁺ cells, and we did not observe a significant increase in CPT1a expression in Mitf^{high} cells within our scRNA-seq data. Here, we believe that similar results can be achieved through the downregulation of MITF. However, implementing shRNA-mediated downregulation of MITF in B16-Mitf cells and conducting subsequent animal experiments would necessitate a longer time frame, and regrettably, we are unable to complete these experiments at this time. In our recent study⁹, we have identified a novel, specific inhibitor of MITF called TT-012. This inhibitor specifically targets dynamic MITF, disrupting dimer formation and DNA-binding capacity. In our forthcoming research, we intend to investigate the role of MITF in FAO using this inhibitor. Your suggestion and understanding in this regard are greatly appreciated.

③The authors propose the hypothesis that MITF_{high} cells are more dependent on FAO?

Response: I apologize for any misunderstanding caused by our previous statement. Our intention was to illustrate that MITF may play a role in mediating LN metastasis through the promotion of FAO. However, we acknowledge that MITF's biological functions are multifaceted and complex, and LN metastasis likely involves various factors beyond MITF. Thank you for pointing out this clarification, and we appreciate your understanding of the nuanced nature of these processes.

④ Could you please discuss this in the light of the following paper: <https://doi.org/10.1016/j.molcel.2019.10.014> ?

Response: The insights from Yurena et al.¹⁰ regarding MITF's influence on fatty acid metabolism and tumor cell proliferation, along with our own scRNA-seq and ST-seq data, significantly enrich our comprehension of MITF's multifaceted functions. Furthermore, the reference to a study highlighting the prevalence of MITF

amplification in metastatic melanoma and its correlation with patient survival underscores the intricate role MITF plays in melanoma progression¹¹. Recognizing the multifaceted nature of MITF's involvement and its potential connection to the promotion of LN metastasis through FAO allows us to offer a more comprehensive interpretation of our research findings. We are sincerely grateful for your thoughtful feedback and guidance throughout this discussion.

⑤ Again, the conclusive summary at the end of Figure 4 results section (line 281) is not convincing, please carefully rephrase: “All of these confirm that MITF promotes LN metastasis via strengthening the FAO pathway in AM.”

Response: I appreciate your understanding. Allow me to rephrase (line 257-259): All of these findings collectively support the notion that MITF contributes to increased FAO activity, thereby promoting LN metastasis in AM.

Figure 5 related

【 Comment 27 】 Early-stage AM lesions show higher amounts of presumably cytotoxic *FGFBP2*⁺ NKT cells, probably a sign of the host's immune response to keep the tumor in check. Later-stage primary AM lesions show fewer of these NKT cells, as they probably exhaust over time and thereby leaving AM melanoma cells behind, which can disseminate to lymph nodes. The NKT observation is interesting as such but its direct implication in LN-metastasis formation is not addressed. So line 348: “NKT key roles in LN metastasis” should be toned down.

Response: Thank you for your thoughtful suggestion. We have incorporated your correction as follows (line 322-323): “These results indicate that decreased *FGFBP2*⁺NKT cells are closely correlated with LN metastasis in AM.” We truly appreciate the reviewer's valuable input, and we are committed to conducting further research to elucidate the intricate mechanisms underlying the role of NKT cells in tumor metastasis. Your feedback has been immensely helpful.

【Comment 28】 Figure 5I&J: Also here remarks about the TCGA SKCM cohort and signature apply, see comments about Figure 4A.

Response: Thank you for your insightful feedback. In alignment with your concerns regarding the issues in **Figure 4A**, particularly the presence of mostly metastatic lesions and non-acral melanoma samples in the TCGA-SKCM data, we have taken the decision to exclude the TCGA-SKCM results (**Figure 5j**).

To address this concern, we conducted an analysis using an RNA-seq dataset comprising 26 primary Acral melanomas (GSE162682)¹. We have incorporated the results into **Supplementary Figure 12f and 12g**. The *FGFBP2*⁺NKT signature we employed in this analysis was derived from our scRNA-seq data and established through a comparison with the remaining 4 AM clusters. The specific gene list utilized can be found in the Source data (**Supplementary Figure 11a-b**). Our threshold for selection was set at $\text{avg_log2FC} > 0.5$ and $\text{adj. p-value} < 0.01$.

We highly appreciate your meticulous attention to detail and your valuable recommendations, which have undoubtedly improved the quality of our research.

【Comment 29】 Figure 5KL: It would be interesting to know more about the AM cohort which was used for NKT staining and quantification. How many different lesions/patients were stained? Do the authors think that para-tumor NKT cells actively play a role in killing tumor cells? When relating high vs. low NKT levels in bulk RNAseq data to OS, a distinction in NKT localization is not possible, which is a limitation.

Response: Thank you for your question. In our study, we employed a paraffin-embedded melanoma tissue microarray comprising 101 paired tumor and

non-tumor Acral melanoma samples to quantify *FGFBP2*⁺NKT cell numbers within each tissue spot through multiplex IHC staining. We employed stringent criteria, counting cells as positive only if they exhibited staining for CD8, NCAM1, and *FGFBP2*. This tissue microarray also included clinical data suitable for overall survival (OS) analysis. We stratified these spots into high and low groups based on *FGFBP2*⁺NKT cell counts and subsequently conducted OS analysis using Kaplan-Meier plots.

Our belief is that *FGFBP2*⁺NKT cells in both the tumor and para-tumor areas possess the same anti-tumor effect. Nevertheless, we observed a significant reduction in *FGFBP2*⁺NKT cells (known for their anti-tumor roles) within the tumor area, which correlated with tumor progression.

Regarding the bulk RNA-seq data, we specifically examined the role of *FGFBP2*⁺NKT cells within the tumor area and elucidated their relationship with patient prognosis.

We genuinely appreciate your question and the opportunity to provide additional clarification on our methodology. Your inquiries help enhance the comprehensibility and validity of our research.

【Comment 30】 Were there any NKT cells in the B16-Mitf syngeneic footpad model?

Response: Thank you for your question. The suggestion to detect *FGFBP2*⁺NKT cells in mouse tumor tissue is indeed valuable for validating our results. We made an earnest effort to detect this specific cell subpopulation, characterized by positivity for CD8, NK1.1, and *FGFBP2* markers. Regrettably, we encountered a challenge in sourcing the *FGFBP2* anti-mouse antibody, which limited our ability to perform this analysis.

However, we took the initiative to consult the existing literature and identified several studies that have successfully confirmed the presence of NKT cells within the tumor microenvironment of B16 tumor models (as evidenced by references¹²⁻¹⁴). These findings lend further support to our research.

Your input and inquiries are immensely appreciated, as they contribute to the

refinement and credibility of our study.

【Comment 31】 Besides NKT cells being present in a higher proportion among early AM lesions, cyt. CD8T cells and NK cells were also slightly increased as one would expect. While Supp Fig9 is showing the UMAP of the different immune cells split by patient, it is difficult to detect the NKT cells. I would appreciate a violin plot representation split by patients (including stats). It seems like patient 2 (LM-) has almost no NKT cells?

Response: Thank you for your question. Your suggestion to provide a more detailed representation of *FGFBP2*⁺NKT cells distribution among patients is valuable. To address this, we have included a histogram in **Figure 5i** (as shown below), which offers a patient-specific breakdown. Specifically, in Patient2, *FGFBP2*⁺NKT cells constitute approximately 8.27% of all T/B/NK cells.

We appreciate your insightful feedback and your contributions to enhancing the clarity and completeness of our research presentation.

【Comment 32】 The Figure 5 legend is confusing: “Impaired anti-tumor immunity, characterized by decreased NKT_*FGFBP2* cells accelerating the LN metastasis in AM”. The term “accelerating” would ask for timing experiments, for example depleting NKT cells and show that LN mets arise faster... Please adapt. To me: Late-stage primary AM lesions exhibit lower NKT cell numbers.

Response: Thank you for your suggestion, and we have corrected as: Decreased *FGFBP2*⁺NKT cells in LN⁺AM.

Figure 6 related

【 Comment 33 】It is unclear, which M1/M2 polarization signatures were used here? It would be fair to mention that the classical way of binary M1/M2 macrophage classification is debated since the recent single-cell based approaches: <https://doi.org/10.1016/j.it.2022.04.008>. I guess Arginase 1 would be a M2 marker here? Also, the Pro and Anti-inflammatory signatures are not further explained (Figure 6H).

Response: Thank you for providing us with the latest literature¹⁵, which reviews recent major studies on single-cell transcriptome, epigenome, metabolome, and spatial omics of cancer, with a specific focus on TAMs. This comprehensive resource has shed light on the diversity of TAM subsets, including IFN-TAMs, Reg-TAMs, Inflam-TAMs, LA-TAMs, Angio-TAMs, RTM-TAMs, and Prolif-TAMs, based on signature genes, enriched pathways, and predicted functions.

We initially used the M1/M2 nomenclature, as it is widely recognized to describe macrophage diversity. Our macrophage signature analysis was based on a recent study published in *CELL*¹⁶, which provided gene signatures for M1/M2 classification. Among these signatures, Arginase 1 is a marker of M2 macrophages. Our findings indicated that most macrophages in our study exhibited a skew toward an M2-like phenotype, with the exception of *APOE*⁺Mac cells, which displayed low M1/M2-like scores. This suggests that the majority of macrophages exert anti-inflammatory effects in the melanoma ecosystem.

We genuinely appreciate the reviewer for sharing the suggested method for regrouping macrophages. In our subsequent work, we will consider adopting this recommended approach to refine our macrophage classification. Once again, we extend our gratitude to the reviewer for their valuable input and guidance.

Figure 7 related

【 Comment 34 】 Figure 7J: It shows expected features of blood vessels (pericytes/mural cells giving stability to endothelium). The increased number of “fibro-vascular niches” in late-stage AM lesions might be just an indicator of better

vascularization of bigger tumors?

Response: Thank you for your question. Through GSVA analysis, we observed that the angiogenesis pathway was enriched in all CAF subclusters, suggesting that these CAF subclusters may contribute to tumor angiogenesis. We confirmed that endothelial cells and fibroblasts exhibited significant co-localization in their spatial distribution. Both CAFs and endothelial cells play pivotal roles in tumor progression. In our study, we also found that there was increased co-localization in LN⁺AM lesions with statistically significant differences. These findings indicate that the fibrovascular niches may indeed be associated with LN metastasis.

We sincerely appreciate your thoughtful question, which has allowed us to clarify and expand upon this aspect of our research. Your insights are highly valuable to our work.

【Comment 35】 This paragraph about the stromal compartment contains many typos and poorly structured sentences.

Response: We sincerely apologize for our oversight. We have corrected the typos and poorly structured sentences in our manuscript. To prevent similar mistakes in the future, we have thoroughly reviewed and edited the entire article. We also invited a native speaker to improve and refine the language issues of the article. We genuinely appreciate your diligence in identifying these errors, and your feedback has been instrumental in improving the quality of our work. Thank you once again for your valuable input.

【Comment 36】 Careful line405: the lymphatic endothelial marker is PROX1 and not PPROX1.

Response: We sincerely apologize for our carelessness. We have made corrections and checked the entire articles to avoid the same mistakes.

Figure 8 related

【Comment 37】“Of note, we found the interaction number and strengths were greater in the ecosystem of LN+ patients compared with that of LN- patients, suggesting that a more complex communication networks in LN+AM”. Can you strengthen this claim statistically?

Response: Thank you for your question. We have carefully examined the data and conducted an analysis of the interaction number and strengths (as shown below). The results indicate a slightly higher interaction number and strengths in the LN⁺AM group compared to the LN⁻AM group. However, it's important to note that this difference did not reach statistical significance, with a p-value of 0.220. As a result, we have revised the conclusion in the manuscript to accurately reflect this finding (line410-412): "Of note, we found that the interaction number and strengths were slightly greater in the ecosystem of LN⁺AM compared to that of LN⁻AM, while there was no significant difference ($P = 0.220$) (**Supplementary Fig. 15c, d**).” We appreciate your insightful question, which has allowed us to clarify this aspect of our research.

【Comment 38】 Careful line 454: CLEC and not CELC.

Response: We genuinely apologize for any oversight on our part. We have taken the necessary steps to correct the errors and have thoroughly reviewed the entire manuscript to prevent similar mistakes in the future. Your diligence in reviewing our work is greatly appreciated, and we are thankful for your valuable feedback.

【Comment 39】 CCL, CXCL, CLEC pathways, were decreased in LN+ patients, which may be related to the more immunosuppressed TME mainly because of a poorer immune environment, I believe.

Response: Thank you for bringing our mistake to our attention. We have addressed this issue by making the necessary correction. Specifically, we have modified the text to read (line 421-423): "Compared to LN⁻AM, the immune/inflammatory pathways, such as the CCL, CXCL, and CLEC signals, were decreased in LN⁺AM, which may be related to a weaker TME." Your input has been invaluable in improving the accuracy of our work, and we are sincerely grateful for your feedback.

【Comment 40】 I appreciate the CellChat based effort to predict cell-cell interactions from scRNA-seq. It would be beneficial for the manuscript to validate at least one receptor-ligand pair using an antibody-based technique (IF, IHC) in the AM TME cohort. Finally, I would like to recommend the author the COMMOT pipeline (<https://github.com/zcang/COMMOT>), which allows inference of signaling directionality in spatial data. It would be tempting to reanalyze the Visium data to map some of these predicted signaling pathways (MK signaling network for example).

Response: We would like to express our gratitude to the reviewer for the valuable suggestion. As per your recommendation, we conducted the multiplex IHC assay to detect the ligand-receptor pairs in the GALECTIN pathway, which includes the LGALS9-CD44, LGALS9-CD45, and LGALS9-HAVCR2 pairs. This allowed us to confirm the interaction of these ligand-receptor pairs in AM, as depicted in **Figure 8f** (as shown below).

In addition, to enhance our analysis of the Visium data and to better visualize some of the predicted signaling pathways, we applied the stlearn pipeline. Through this process, we were able to validate the spatial distribution of ligand-receptor pairs within the MK and GALECTIN pathways, as illustrated in **Figure 8e and**

Supplementary Figure 16g.

We sincerely appreciate your guidance and insights, which have significantly contributed to the improvement of our research.

Reviewer #4 (Remarks to the Author): Expert in single-cell and spatial transcriptomics, and computational genomics

Authors use scRNA-seq and spatial transcriptomics technologies to explore the tumor ecosystem of AM with different LN metastasis status, and the dynamic changes during tumor early dissemination at spatial and temporal levels. The biological findings they found through public datasets were also verified through a large number of clinical datasets which could contribute to a better understanding of LN metastasis and novel therapeutic strategies for early dissemination of AM. Authors conducted comprehensive experiments to make the findings reliable and convincing.

There are some issues and comments authors should consider:

Response: We want to express our sincere gratitude for your insightful comments on our manuscript. Your suggestions have been invaluable in refining our manuscript and will undoubtedly contribute to the advancement of our ongoing research. We have diligently reviewed and incorporated your feedback, and we sincerely hope that the revisions align with your expectations and standards. Your time and expertise are greatly appreciated.

【Comment 1】 I noticed that authors used the resolution as 2.0 in the scRNA-seq data clustering and 38 cell clusters were obtained. From my knowledge, 2.0 is a kind of high resolution for cell clustering which would separate some clusters into multiple adjacent sub-clusters forcibly without biological meanings. For example, Figure 1B and D showed that Treg, T cell and NK are pretty close in UMAP and they share multiple similar marker genes. I just wonder if using lower resolution could get similar results which would also help verify the robustness of author's results.

Response: Regarding the reviewer's concern, we would like to express our gratitude for the valuable feedback. We have performed cell clustering using a resolution of 0.5 with the scRNA-seq data. This analysis resulted in a total of 19 sub-clusters, which we have meticulously annotated into 12 distinct cell types. These findings are in alignment with the outcomes obtained at a resolution of 2.0, as previously presented. Your keen observation and input are greatly appreciated.

【Comment 2】 Please add the reference of Seurat's paper in line 101.

Response: Thank you for your constructive suggestion. We have taken your advice and added the reference to Seurat's paper (line 662). Your input is much appreciated.

【Comment 3】 From the recent benchmarking paper on spatial transcriptomics deconvolution task (<https://doi.org/10.1038/s41467-023-37168-7>), there are several better choices than RCTD authors used, such as Cell2location and CARD. Authors could read this benchmarking paper and use at least one more popular deconvolution method to verify the reliability of deconvolution results by using authors' spatial

transcriptomics data.

Response: Thanks for the reviewer's very meaningful recommendation. Here, we have also applied the CARD deconvolution method to delineate the 13 cell types annotated in the scRNA-seq data (**Fig. 2**, as shown below). The results were found to be consistent with those obtained using the RTCD method (shown in **Supplementary Fig. 8b**, as shown below), reinforcing the reliability of our deconvolution results. Your valuable input is greatly appreciated.

[Comment 4] Authors could discuss the exploration of heterogenous cancer tissue at the 3D level through spatial transcriptomics technologies. I believe it would be an

interesting direction to explore and understand cancer.

Response: Thanks for the reviewer's very meaningful suggestion. We firmly believe that exploring cancer heterogeneity at the 3D level is an interesting and promising direction. Such an approach would not only expand downstream analysis possibilities but also provide unparalleled depth in investigating cell types and cell-cell interactions within the tumor microenvironment. We have incorporated this important consideration into the discussion section of the manuscript (line 553-558). Additionally, we are eager to leverage 3D spatial analysis techniques in future investigations of Acral Melanoma.

"Third, our study currently lacks three-dimensional (3D) spatial analysis. Tumors grow within a 3D environment, and our current presentation is limited to 2D spatial information concerning the interactions between tumors and their TME. We recognize the significance of considering the 3D context of tumor development. Advanced methodologies such as PASTE can be instrumental in expanding our ability to perform downstream analysis and elucidating cellular interaction networks within the intricate 3D spatial structures of tumors."

【Comment 5】 Authors should supply the reference paper or resource about how they annotate the clusters to corresponding cell types through the marker genes.

Response: Thanks for your valuable suggestion. The marker genes we employed for annotating the clusters to their corresponding cell types were primarily presented in **Figure 1d**. We have also provided the references^{2,3,17} for these marker genes below and cited them appropriately in the manuscript.

"A single-cell analysis reveals tumor heterogeneity and immune environment of acral melanoma." *Nat Commun.* 2022 Nov 25;13(1):7250. doi: 10.1038/s41467-022-34877-3.

"Single-cell Characterization of the Cellular Landscape of Acral Melanoma Identifies Novel Targets for Immunotherapy." *Clin Cancer Res.* 2022 May 13;28(10):2131-2146. doi: 10.1158/1078-0432.

"Single-cell transcriptomic analysis suggests two molecularly subtypes of

intrahepatic cholangiocarcinoma." Nat Commun. 2022 Mar 28;13(1):1642. doi: 10.1038/s41467-022-29164-0.

We appreciate your diligent review of our manuscript.

【Comment 6】 If possible, authors could use more advanced ST technologies with higher spot resolution (such as stereo-seq) to explore the cancer which could give a more fine-grained picture of tissue. Although 10X Visium is a popular commercial technology, the resolution is too low (50 μm of diameter per spot, 10-20 cells contained in a spot).

Response: Thank you so much for your valuable suggestion, and we truly appreciate your perspective. We couldn't agree with you more on the potential of Stereo-seq. However, it's worth noting that when we initiated this project, there were no published studies available on Stereo-seq, which is why we opted to use 10X Visium for the spatial transcriptome analysis.

Coincidentally, we have recently established a collaboration with BGI-Southwest (Chongqing, China) to undertake a Stereo-seq project focused on melanoma LN metastasis. We believe this upcoming project will serve as a valuable supplement and continuation of our current study.

Once again, we sincerely thank you for your thoughtful insights and suggestions. Your feedback has been instrumental in shaping our research direction.

【Comment 7】 Could authors explain which findings are first discovered by them, and which findings are proposed by previous works and authors used scRNA-seq & ST data to verify them? It is important to show the novelty of this paper.

Response: We want to express our sincere gratitude for your valuable feedback. It has been incredibly insightful, and we apologize for any shortcomings in conveying the innovation points of our study. To address this, we would like to highlight the key novelties in our research, as they are discussed in detail in the manuscript's discussion section.

Firstly, our study confirms the well-documented observation that Acral Melanom

exhibits a highly immunosuppressive TME compared to Cutaneous Melanoma. Furthermore, we provide evidence that AM with lymph node metastasis (LN⁺AM) displays an even more immunosuppressive TME than AM without metastasis (LN⁻AM). We substantiate this finding through comprehensive analysis using both single-cell RNA sequencing (scRNA-seq) and spatial transcriptomics (ST-seq) data.

Secondly, we identify and characterize two crucial cell populations, *MITF*⁺MEL and *FGFBP2*⁺NKT cells, that are closely associated with LN metastasis and unfavorable prognosis in AM patients. We validate these findings not only within our AM patient cohort but also through external data sources, strengthening the clinical relevance of our discoveries.

Thirdly, we contribute to the understanding of fatty acid oxidation (FAO) pathway activation in primary melanoma lesions that are predisposed to metastasize. Unlike previous studies focusing solely on metastatic lesions, our research highlights the early activation of the FAO pathway in primary lesions, shedding light on its significance in tumor progression.

Last but certainly not least, our study implicates MITF as a key factor in promoting LN metastasis in primary tumors by enhancing the FAO pathway. This connection adds a crucial layer of insight into the mechanisms underlying AM metastasis.

Your guidance has been invaluable in refining the clarity and emphasis of these novel aspects of our work. We genuinely appreciate your contributions to improving our manuscript.

【Comment 8】 The last comma should be changed to full stop in line 189.

Response: We want to extend our deepest gratitude for your careful review and valuable feedback. We sincerely apologize for any carelessness in our previous submissions. To prevent any recurrence of these errors, we have diligently corrected them and conducted a thorough review of the entire manuscript. Your dedication to ensuring the accuracy and quality of our work is greatly appreciated. Thank you once again for your insightful observations and contributions to improving our research.

References

1. Farshidfar, F. et al. Integrative molecular and clinical profiling of acral melanoma links focal amplification of 22q11.21 to metastasis. *Nat. Commun.* **13**, 898 (2022).
2. Li, J. et al. Single-cell Characterization of the Cellular Landscape of Acral Melanoma Identifies Novel Targets for Immunotherapy. *Clin. Cancer Res.* **28**, 2131-2146 (2022).
3. Zhang, C. et al. A single-cell analysis reveals tumor heterogeneity and immune environment of acral melanoma. *Nat. Commun.* **13**, 7250 (2022).
4. Genomic Classification of Cutaneous Melanoma. *Cell.* **161**, 1681-1696 (2015).
5. Cirenajwis, H. et al. Molecular stratification of metastatic melanoma using gene expression profiling: Prediction of survival outcome and benefit from molecular targeted therapy. *Oncotarget.* **6**, 12297-12309 (2015).
6. Belote, R. L. et al. Human melanocyte development and melanoma dedifferentiation at single-cell resolution. *Nat. Cell Biol.* **23**, 1035-1047 (2021).
7. Stuart, T. et al. Comprehensive Integration of Single-Cell Data. *Cell.* **177**, 1888-1902 (2019).
8. McGinnis, C. S., Murrow, L. M. & Gartner, Z. J. DoubletFinder: Doublet Detection in Single-Cell RNA Sequencing Data Using Artificial Nearest Neighbors. *Cell Syst.* **8**, 329-337 (2019).
9. Liu, Z. et al. A unique hyperdynamic dimer interface permits small molecule perturbation of the melanoma oncoprotein MITF for melanoma therapy. *Cell Res.* **33**, 55-70 (2023).
10. Vivas-Garcia, Y. et al. Lineage-Restricted Regulation of SCD and Fatty Acid Saturation by MITF Controls Melanoma Phenotypic Plasticity. *Mol. Cell.* **77**, 120-137 (2020).
11. Garraway, L. A. et al. Integrative genomic analyses identify MITF as a lineage survival oncogene amplified in malignant melanoma. *Nature.* **436**, 117-122 (2005).
12. Liu, Y. et al. 30-color full spectrum flow cytometry panel for deep

- immunophenotyping of T cell subsets in murine tumor tissue. *J. Immunol. Methods.* **516**, 113459 (2023).
13. Liu, X. et al. NK and NKT cells have distinct properties and functions in cancer. *Oncogene.* **40**, 4521-4537 (2021).
 14. Seitz, C. et al. Tumor Cell-Based Vaccine Generated With High Hydrostatic Pressure Synergizes With Radiotherapy by Generating a Favorable Anti-tumor Immune Microenvironment. *Front. Oncol.* **9**, 805 (2019).
 15. Ma, R. Y., Black, A. & Qian, B. Z. Macrophage diversity in cancer revisited in the era of single-cell omics. *Trends Immunol.* **43**, 546-563 (2022).
 16. Azizi, E. et al. Single-Cell Map of Diverse Immune Phenotypes in the Breast Tumor Microenvironment. *Cell.* **174**, 1293-1308 (2018).
 17. Song, G. et al. Single-cell transcriptomic analysis suggests two molecularly subtypes of intrahepatic cholangiocarcinoma. *Nat. Commun.* **13**, 1642 (2022).

REVIEWERS' COMMENTS

Reviewer #1 (Remarks to the Author):

I appreciate the authors' efforts in addressing all my original comments. I am happy with their responses regarding my questions 1-3. I also appreciate their explanation regarding point 4, about the mouse experiment with the tumour in the foot pad. However, I would still ask the editors and authors to re-review that these are indeed within the limits established by ethics - the 2000m³ limit is, as far as I know, for growth in the back or other body sites that are not the foot pad.

Reviewer #3 (Remarks to the Author):

In my opinion, the authors have substantially improved the manuscript by addressing most of my concerns sufficiently. I am still convinced that this study provides an important resource for studying acral melanoma at single cell resolution.

Nonetheless, before recommending this article for publication in Nat.Com., I would like to invite the authors to discuss the following point in greater detail (Discussion section of the paper):

It is surprising that their malignant MITF+MEL population in AM is more similar to the mesenchymal-like malignant cluster in CM. This is a noteworthy discrepancy as the mesenchymal and neural-crest-like CM melanoma states are generally more dedifferentiated, characterized by lower MITF (regulon) expression. Another data set to interrogate and compare with would be Tsoi et al. (PMID: 29657129). Whereas MITF_{low} malignant melanoma cell states are associated with disease progression and therapy-resistance in CM, do we expect a mirrored scenario in AM?

Reviewer #4 (Remarks to the Author):

After the review, the authors have fully solved my concerns and supplied sufficient revisions to the new version of the manuscript. I recommend to accept this paper.

Reviewer #1 (Remarks to the Author):

I appreciate the authors' efforts in addressing all my original comments. I am happy with their responses regarding my questions 1-3. I also appreciate their explanation regarding point 4, about the mouse experiment with the tumor in the foot pad. However, I would still ask the editors and authors to re-review that these are indeed within the limits established by ethics - the 2000mm³ limit is, as far as I know, for growth in the back or other body sites that are not the foot pad.

Response: We would like to express our sincere gratitude to the reviewer for raising the question regarding tumor size/burden. We would also like to clarify the ethical considerations and provide further explanations for certain aspects of our study.

Firstly, we want to emphasize that this animal experiment was conducted in accordance with the guidelines and regulations set by the ethical review board of Zhongshan Hospital, Fudan University.

Our ethical review board stipulates that the maximum diameter of tumors should not exceed 2 cm. However, there is no specific regulation regarding tumor volume for the foot pad model, as it is a relatively less commonly used model in research. Therefore, neither our ethical institution nor relevant literature have provided specific guidelines on tumor size and volume for this particular model.

In this study, we strictly adhered to the principles of the 3Rs (Replacement, Reduction, and Refinement) and ensured that the diameter of all tumors did not exceed 2 cm, in accordance with the ethical guidelines. It is worth noting that three "tumors" had a volume exceeding 2000 mm³, but this is because the volume includes not only the tumor itself but also the volume of the mouse paws. To avoid any confusion, we have modified the vertical axis of Fig. 4j from "tumor volume" to "foot pad volume".

Fig. 4g was designed to demonstrate a typical lymph node metastasis, which is why the tumor burden appears larger in this particular mouse. However, to eliminate any ambiguity, we intend to replace it with a typical size of the primary lesion.

In order to achieve a higher lymph node (LN) positive rate, especially considering that the LN metastasis rate is known to be lower after Etomoxir treatment¹, we chose to extend the study endpoint to ensure a sufficient LN positive rate for statistical analysis, and this was approved by our ethical review board. However, even with this extension, the LN positive rate at the endpoint of our study was only 59% (16/27), and only 22% (2/9) of mice in the Etomoxir treatment group experienced lymph node metastasis.

The present of ulcers is primarily attributed to poor blood supply to the limbs, which is consistent with clinical observations as patients with acral melanoma often have a higher incidence of ulcers.

Once again, we would like to express our deep appreciation to the reviewer for their insightful guidance, which has greatly contributed to the refinement of our study.

Reviewer #3 (Remarks to the Author):

In my opinion, the authors have substantially improved the manuscript by addressing most of my concerns sufficiently. I am still convinced that this study provides an important resource for studying acral melanoma at single cell resolution.

Nonetheless, before recommending this article for publication in Nat.Com., I would like to invite the authors to discuss the following point in greater detail (Discussion section of the paper):

It is surprising that their malignant MITF⁺MEL population in AM is more similar to the mesenchymal-like malignant cluster in CM. This is a noteworthy discrepancy as the mesenchymal and neural-crest-like CM melanoma states are generally more dedifferentiated, characterized by lower MITF (regulon) expression. Another data set to interrogate and compare with would be Tsoi et al. (PMID: 29657129). Whereas MITF_{low} malignant melanoma cell states are associated with disease progression and therapy-resistance in CM, do we expect a mirrored scenario in AM?

Response: We would like to express our sincere appreciation to the reviewer for raising this question.

MITF is an oncogene that plays a crucial role in tumor initiation, progression, and relapse in melanoma. In our study, we found that the MITF⁺MEL subcluster was functionally enriched in mesenchymal, stem cell, and neural crest development in AM, which is typically associated with lower levels or activity of MITF in CM. There are several reasons that can explain this finding.

Firstly, upon careful examination of our MITF⁺MEL cluster, we observed that this subgroup also expresses high-level genes such as *MYC*, *SNAI2*, *KIT*, and *ERBB3*, which are closely associated with tumor invasion and metastasis. This similarity in gene expression profiles suggests that the biological behavior of this subgroup is more similar to the mesenchymal-like malignant clusters in CM.

Secondly, it is important to acknowledge the significant differences between the two subtypes of melanoma, AM and CM, including gene mutation map, tumor

microenvironment, and therapeutic efficacy. These differences can be attributed to subtype variations and inherent tumor heterogeneity.

Thirdly, we believe that the involvement of MITF in melanoma is multifaceted and complex, and its functions may vary depending on the temporal and spatial context of the tumor environment. For instance, low levels of MITF can lead to G1-arrested, invasive, and senescent cells, while cells expressing MITF can either proliferate or differentiate. Additionally, MITF amplification has been found to be more prevalent in metastatic disease and correlated with decreased overall survival rates in melanoma patients.

Finally, and most importantly, unlike previous studies that compared paired metastatic and primary tumor lesions, our research primarily focused on comparing two types of primary tumor lesions: those without lymph node metastasis and those with lymph node metastasis. The purpose was to identify the cell population that plays a crucial role in the early metastasis of primary tumors, specifically tumor cells that are about to metastasize but have not yet done so, and are in an intermediate state between primary and metastatic lesions. In this special state, we demonstrated that MITF is also involved in regulating fatty acid metabolism, which is reported by another study that MITF is a lineage-specific regulator of the fatty acid desaturase SCD². The overactivated FAO pathway promotes the metastatic potential of tumor cells¹. Therefore, we concluded that in the early metastatic stages, MITF can increase the LN metastasis ability of AM cells by enhancing FAO activity.

Here, in order to better describe the cell subgroup involved in tumor invasion and metastasis, we renamed it the *MYC*⁺MEL subgroup as *MYC* is also a significantly overexpressed gene in this subgroup.

However, our study did not include analysis of lymph node metastasis and other distant metastases, and we lacked in-depth analysis of MITF function in these metastatic lesions. In future work, we plan to refine this aspect and comprehensively analyze the dynamic process of tumor cell metastasis from the primary lesion to lymph nodes and other organs. Nonetheless, our study has provided valuable insights into the intermediate state of MITF function between primary and metastatic lesions,

specifically how MITF can enhance metastatic potential by regulating energy metabolism.

Your insightful inquiry has significantly contributed to clarifying the characterization of MITF function and the *MITF*⁺MEL subcluster. We are truly grateful for your engagement in advancing the depth and accuracy of our research.

Reviewer #4 (Remarks to the Author):

After the review, the authors have fully solved my concerns and supplied sufficient revisions to the new version of the manuscript. I recommend to accept this paper.

Response: Thank you for your comments and support for this study.

References

1. Lee, C. K. et al. Tumor metastasis to lymph nodes requires YAP-dependent metabolic adaptation. *Science*. **363**, 644-649 (2019).
2. Vivas-Garcia, Y. et al. Lineage-Restricted Regulation of SCD and Fatty Acid Saturation by MITF Controls Melanoma Phenotypic Plasticity. *Mol. Cell*. **77**, 120-137 (2020).